# Relationship between Batch Size and Number of Steps Needed for Nonconvex Optimization of Stochastic Gradient Descent using Armijo-Line-Search Learning Rate

**Yuki Tsukada**                                                                                   *yuki.t.1119@iiduka.net*
*Department of Computer Science*
*Meiji University*

**Hideaki Iiduka**                                                                                  *iiduka@cs.meiji.ac.jp*
*Department of Computer Science*
*Meiji University*

**Reviewed on OpenReview:** *https://openreview.net/forum?id=pqZ6nOm3WF*

## Abstract

While stochastic gradient descent (SGD) can use various learning rates, such as constant or diminishing rates, previous numerical results showed that SGD performs better than other deep-learning optimizers when it uses learning rates given by line search methods. In this paper, we perform a convergence analysis on SGD with a learning rate given by an Armijo line search for nonconvex optimization indicating that the upper bound of the expectation of the squared norm of the full gradient becomes small when the number of steps and the batch size are large. Next, we show that, for SGD with the Armijo-line-search learning rate, the number of steps needed for nonconvex optimization is a monotone decreasing convex function of the batch size; that is, the number of steps needed for nonconvex optimization decreases as the batch size increases. Furthermore, we show that the stochastic first-order oracle (SFO) complexity, which is the stochastic gradient computation cost, is a convex function of the batch size; that is, there exists a critical batch size that minimizes the SFO complexity. Finally, we provide numerical results that support our theoretical results.

## 1 Introduction

### 1.1 Background

Nonconvex optimization is useful for training deep neural networks, since the loss functions called the expected risk and empirical risk are nonconvex and they need only be minimized in order to find the model parameters. Deep-learning optimizers have been presented for minimizing the loss functions. The simplest one is stochastic gradient descent (SGD) (Robbins & Monro, 1951; Zinkevich, 2003; Nemirovski et al., 2009; Ghadimi & Lan, 2012; 2013) and there are numerous theoretical analyses on using SGD for nonconvex optimization (Jain et al., 2018; Vaswani et al., 2019; Fehrman et al., 2020; Chen et al., 2020; Scaman & Malherbe, 2020; Loizou et al., 2021; Umeda & Iiduka, 2025). Variants of SGD have also been presented, such as momentum methods (Polyak, 1964; Nesterov, 1983) and adaptive methods including Adaptive Gradient (AdaGrad) (Duchi et al., 2011), Root Mean Square Propagation (RMSProp) (Tieleman & Hinton, 2012), Adaptive Moment Estimation (Adam) (Kingma & Ba, 2015), Adaptive Mean Square Gradient (AMSGrad) (Reddi et al., 2018), and Adam with decoupled weight decay (AdamW) (Loshchilov & Hutter, 2019). SGD and its variants are useful for training not only deep neural networks but also generative adversarial networks (Heusel et al., 2017; Naganuma & Iiduka, 2023; Sato & Iiduka, 2023).

The performance of deep-learning optimizers for nonconvex optimization depends on the batch size. The previous numerical results in (Shallue et al., 2019) and (Zhang et al., 2019) have shown that the number of

steps $K$ needed to train a deep neural network halves for each doubling of the batch size $b$ and that there is a region of diminishing returns beyond the *critical batch size* $b^\star$. This fact can be expressed as follows: there is a positive number $C$ such that $N := Kb \approx C$ for $b \leq b^\star$ and $N := Kb \geq C$ for $b \geq b^\star$. The deep neural network model uses $b$ gradients of the loss functions per step. Hence, when $K$ is the number of steps required to train a deep neural network, the model has a stochastic gradient computation cost of $Kb$. We will define the *stochastic first-order oracle (SFO) complexity* (Iiduka, 2022; Sato & Iiduka, 2023) of a deep-learning optimizer to be $N := Kb$. From the previous numerical results in (Shallue et al., 2019) and (Zhang et al., 2019), the SFO complexity is minimized at a critical batch size $b^\star$ and there are diminishing returns once the batch size exceeds $b^\star$. Therefore, it is desirable to use the critical batch size when minimizing the SFO complexity of the deep-learning optimizer.

Not only the batch size but also the learning rate affects the performance of deep-learning optimizers for nonconvex optimization. A performance measure of a deep-learning optimizer generating a sequence $(\boldsymbol{\theta}_k)_{k \in \mathbb{N}}$ is the expectation of the squared norm of the gradient of a nonconvex loss function $f$, denoted by $\mathbb{E}[\|\nabla f(\boldsymbol{\theta}_k)\|^2]$. If this performance measure becomes small when the number of steps $k$ is large, the deep-learning optimizer approximates a local minimizer of $f$. For example, let us consider the problem of minimizing a smooth function $f$ (see Section 2.1 for the definition of smoothness). Here, SGD uses a constant learning rate $\alpha = O(\frac{1}{L})$ satisfying $\min_{k \in [K]} \mathbb{E}\left[\|\nabla f(\boldsymbol{\theta}_k)\|^2\right] = O(\frac{1}{K} + \frac{\alpha}{b})$, where $L$ is the Lipschitz constant of $\nabla f$, $b$ is the batch size, and $[K] := \{1, 2, \ldots, K\}$ (see also Table 1). Moreover, SGD using a learning rate satisfying the Armijo condition was presented in (Vaswani et al., 2019). The *Armijo line search* (Nocedal & Wright, 2006, Chapter 3.1) is a standard method for finding an appropriate learning rate $\alpha_k$ giving a sufficient decrease in $f$, i.e., $f(\boldsymbol{\theta}_{k+1}) < f(\boldsymbol{\theta}_k)$ (see Section 2.3.1 for the definition of the Armijo condition).

## 1.2 Motivation

The numerical results in (Vaswani et al., 2019) indicated that using the Armijo-line-search learning rate is superior to using a constant learning rate when using SGD to train deep neural networks in the sense of minimizing the training loss and improving test accuracy. Motivated by the useful numerical results in (Vaswani et al., 2019), we decided to perform convergence analyses on SGD with the Armijo-line-search learning rate for nonconvex optimization in deep neural networks.

Theorem 3 in (Vaswani et al., 2019) is a convergence analysis of SGD with the Armijo-line-search learning rate for nonconvex optimization under a strong growth condition that implies the interpolation property. Here, let $f \colon \mathbb{R}^d \to \mathbb{R}$ be an empirical risk defined by $f(\boldsymbol{\theta}) := \frac{1}{n} \sum_{i \in [n]} f_i(\boldsymbol{\theta})$, where $n$ is the number of training data and $f_i \colon \mathbb{R}^d \to \mathbb{R}$ is a loss function corresponding to the $i$-th training data $z_i$. We say that $f$ has the interpolation property if $\nabla f(\boldsymbol{\theta}) = \mathbf{0}$ implies $\nabla f_i(\boldsymbol{\theta}) = \mathbf{0}$ ($i \in [n]$). The interpolation property holds for optimization of a linear model with the squared hinge loss for binary classification on linearly separable data (Vaswani et al., 2019, Section 2). However, the strong growth condition would be unrealistic for deep neural networks, since their loss functions are nonconvex. The motivation behind this work is thus to show that SGD with the Armijo-line-search learning rate can solve nonconvex optimization problems in deep neural networks.

As indicated in the second paragraph in Section 1.1, the batch size has a significant effect on the performance of SGD. Hence, in accordance with the first motivation stated above, we decided to investigate appropriate batch sizes for SGD with the Armijo-line-search learning rate. In particular, we are interested in verifying whether a critical batch size $b^\star$ minimizing the SFO complexity $N$ exists for training deep neural networks with SGD using the Armijo condition in theory and in practice. This is because the previous studies in (Shallue et al., 2019; Zhang et al., 2019; Iiduka, 2022; Sato & Iiduka, 2023) showed the existence of critical batch sizes for training deep neural networks or generative adversarial networks with optimizers with constant or diminishing learning rates and without Armijo-line-search learning rates.

### 1.3 Contribution

#### 1.3.1 Convergence analysis of SGD with Armijo-line-search learning rates

The first contribution of this paper is to present a convergence analysis of SGD with Armijo-line-search learning rates for general nonconvex optimization (Theorem 3.1); in particular, it is shown that SGD with this rate $\alpha_k$ satisfies that, for all $K \geq 1$,

$$
\min_{k \in [0:K-1]} \mathbb{E}\left[\|\nabla f(\boldsymbol{\theta}_k)\|^2\right] \leq \underbrace{\overbrace{\frac{2(f(\boldsymbol{\theta}_0) - f_*)}{\tilde{\alpha} - (L_n \overline{\alpha} - 1)\overline{\alpha}}}^{C_1} \frac{1}{K}}_{B(\boldsymbol{\theta}_0, K)} + \underbrace{\overbrace{\frac{L_n \overline{\alpha}^2 \sigma^2}{\tilde{\alpha} - (L_n \overline{\alpha} - 1)\overline{\alpha}}}^{C_2} \frac{1}{b}}_{V(\sigma^2, b)},
\tag{1}
$$

where the parameters are defined in Table 1 (see also Theorem 3.1). The inequality (1) indicates that the upper bound of the performance measure $\min_{k \in [0:K-1]} \mathbb{E}[\|\nabla f(\boldsymbol{\theta}_k)\|^2]$ that consists of a bias term $B(\boldsymbol{\theta}_0, K)$ and variance term $V(\sigma^2, b)$ becomes small when the number of steps $K$ is large and the batch size $b$ is large. Therefore, it is desirable to set $K$ large and $b$ large so that Algorithm 1 will approximate a local minimizer of $f$.

The essential lemma to proving (1) is the guarantee of the existence of a lower bound on the learning rates satisfying the Armijo condition (Lemma 2.1). Although, in general, learning rates satisfying the Armijo condition do not have any lower bound (Lemma 2.1(i)), the corresponding learning rates computed by a backtracking line search (Algorithm 1) have a lower bound (Lemma 2.1(ii)). In addition, the descent lemma (i.e., $f(\boldsymbol{y}) \leq f(\boldsymbol{x}) + \langle \nabla f(\boldsymbol{x}), \boldsymbol{y} - \boldsymbol{x} \rangle + \frac{L_n}{2}\|\boldsymbol{y} - \boldsymbol{x}\|^2$ $(\boldsymbol{x}, \boldsymbol{y} \in \mathbb{R}^d)$) holds from the smoothness condition on $f$. Thus, we can prove (1) by using the existence of a lower bound on the learning rates satisfying the Armijo condition and the descent lemma (see Appendix A.2 for details of the proof of Theorem 3.1).

Table 1: Relationship between batch size $b$ and number of steps $K$ to achieve an $\epsilon$–approximation defined by $\min_{k \in [0:K-1]} \mathbb{E}[\|\nabla f(\boldsymbol{\theta}_k)\|^2] \leq \frac{C_1}{K} + \frac{C_2}{b} = \epsilon^2$ for SGD with a constant learning rate $\alpha \in (0, \frac{2}{L_n})$ and for SGD with Armijo-line-search learning rate $\alpha_k \in [\underline{\alpha}, \overline{\alpha}]$ ($[0 : K - 1] := \{0, 1, \ldots, K - 1\}$, $f := \frac{1}{n}\sum_{i \in [n]} f_i$ is bounded below by $f_*$, $L_i$ is the Lipschitz constant of $\nabla f_i$, $L_n := \frac{1}{n}\sum_{i \in [n]} L_i$, $\tilde{\alpha} := \frac{2\delta(1-c)}{L_n}$, $\delta \in (\frac{1}{4}, 1)$, $c \in (0, 1 - \frac{1}{4\delta})$, and $\sigma^2$ is the upper bound of the variance of the stochastic gradient)

| Learning Rate | Upper Bound $\frac{C_1}{K} + \frac{C_2}{b}$ | Steps $K$ | SFO $N$ | Critical Batch $b^\star$ |
|---|---|---|---|---|
| Constant $\alpha \in \left(0, \frac{2}{L_n}\right)$ | $C_1 = \frac{2(f(\boldsymbol{\theta}_0) - f_*)}{(2 - L_n\alpha)\alpha}$ $C_2 = \frac{L_n\alpha\sigma^2}{2 - L_n\alpha}$ | $K = \frac{C_1 b}{\epsilon^2 b - C_2}$ | $N = \frac{C_1 b^2}{\epsilon^2 b - C_2}$ | $b^\star = \frac{2C_2}{\epsilon^2}$ |
| Armijo $\overline{\alpha} \in \left(\frac{1}{L_n}, \frac{2}{L_n}\right)$ | $C_1 = \frac{2(f(\boldsymbol{\theta}_0) - f_*)}{\tilde{\alpha} - (L_n\overline{\alpha} - 1)\overline{\alpha}}$ $C_2 = \frac{L_n\overline{\alpha}^2\sigma^2}{\tilde{\alpha} - (L_n\overline{\alpha} - 1)\overline{\alpha}}$ | $K = \frac{C_1 b}{\epsilon^2 b - C_2}$ | $N = \frac{C_1 b^2}{\epsilon^2 b - C_2}$ | $b^\star = \frac{2C_2}{\epsilon^2}$ |

#### 1.3.2 Steps needed for $\epsilon$–approximation of SGD with Armijo line-search-learning rates

The previous results in (Shallue et al., 2019; Zhang et al., 2019; Iiduka, 2022; Sato & Iiduka, 2023) indicated that, for optimizers, the number of steps $K$ needed to train a deep neural network or generative adversarial networks decreases as the batch size increases. The second contribution of this paper is to show that, for SGD with the Armijo-line-search learning rate, the number of steps $K$ needed for nonconvex optimization decreases as the batch size increases. Let us consider the case in which the right-hand side of (1) is equal to $\epsilon^2$, where $\epsilon > 0$ is the precision. Then, $K$ is a rational function defined for a batch size $b$ by

$$
K = K(b) = \frac{C_1 b}{\epsilon^2 b - C_2},
\tag{2}
$$

where $C_1$ and $C_2$ are the positive constants defined in (1) (see also Table 1). We can easily show that $K$ defined above is a monotone decreasing and convex function with respect to $b$ (Theorem 3.2). Accordingly, the number of steps needed for nonconvex optimization decreases as the batch size increases.

### 1.3.3 Critical batch size minimizing SFO complexity of SGD with Armijo-line-search learning rates

Using $K$ defined by (2) above, we can further define the SFO complexity $N$ of SGD with Armijo-line-search learning rates (see also Table 1):

$$N = Kb = K(b)b = \frac{C_1 b^2}{\epsilon^2 b - C_2}. \tag{3}$$

We can easily show that $N$ is convex with respect to $b$ and that a global minimizer

$$b^\star = \frac{2C_2}{\epsilon^2} = \frac{2L_n \overline{\alpha}^2 \sigma^2}{\{\tilde{\alpha} - (L_n \overline{\alpha} - 1)\overline{\alpha}\}\epsilon^2} \tag{4}$$

exists for it (Theorem 3.3). Accordingly, there is a critical batch size $b^\star$ at which $N$ is minimized.

Here, we compare the number of steps $K_{\rm C}$ and the SFO complexity $N_{\rm C}$ for SGD using a constant learning rate $\alpha$ with $K_{\rm A}$ and $N_{\rm A}$ for SGD using Armijo-line-search learning rate $\alpha_k$ ($\in [\underline{\alpha}, \overline{\alpha}]$). Let $C_{1,{\rm C}}$ (resp. $C_{2,{\rm C}}$) be $C_1$ (resp. $C_2$) in Table 1 for SGD using a constant learning rate and let $C_{1,{\rm A}}$ (resp. $C_{2,{\rm A}}$) be $C_1$ (resp. $C_2$) in Table 1 for SGD using the Armijo-line-search learning rate. We have that

$$\begin{aligned} C_{1,{\rm A}} < C_{1,{\rm C}} \text{ iff } (2 - L_n \alpha)\alpha < \tilde{\alpha} - (L_n \overline{\alpha} - 1)\overline{\alpha}, \\ C_{2,{\rm A}} < C_{2,{\rm C}} \text{ iff } \frac{\overline{\alpha}^2 \sigma_{\rm A}^2}{\tilde{\alpha} - (L_n \overline{\alpha} - 1)\overline{\alpha}} < \frac{\alpha \sigma_{\rm C}^2}{2 - L_n \alpha}, \end{aligned} \tag{5}$$

where $\sigma_{\rm C}^2$ (resp. $\sigma_{\rm A}^2$) denotes the upper bound of the variance of the stochastic gradient for SGD using a constant learning rate $\alpha$ (resp. the Armijo-line-search learning rate). If (5) holds, then SGD using the Armijo-line-search learning rate converges faster than SGD using a constant learning rate in the sense that $\frac{C_{1,{\rm A}} b}{\epsilon^2 b - C_{2,{\rm A}}} = K_{\rm A} < K_{\rm C} = \frac{C_{1,{\rm C}} b}{\epsilon^2 b - C_{2,{\rm C}}}$ and $\frac{C_{1,{\rm A}} b^2}{\epsilon^2 b - C_{2,{\rm A}}} = N_{\rm A} < N_{\rm C} = \frac{C_{1,{\rm C}} b^2}{\epsilon^2 b - C_{2,{\rm C}}}$. It would be difficult to check exactly that (5) holds before implementing SGD, since (5) involves unknown parameters, such as $L_n = \frac{1}{n} \sum_{i \in [n]} L_i$, $\sigma_{\rm C}^2$, and $\sigma_{\rm A}^2$. However, it can be expected that (5) holds, since it is known empirically (Vaswani et al., 2019, Figure 5) that the relationship between the Armijo-line-search learning rate $\alpha_k$ and a constant learning rate $\alpha$ is $\alpha < \alpha_k < \overline{\alpha}$.

### 1.3.4 Numerical results supporting our theoretical results

The numerical results in (Vaswani et al., 2019) showed that SGD with the Armijo-line-search learning rate performs better than other optimizers in training deep neural networks (DNNs). Hence, we seek to determine whether the numerical results match our theoretical results (Sections 1.3.1, 1.3.2, and 1.3.3). We trained residual networks (ResNets) on the CIFAR-10 and MNIST datasets. We numerically found that increasing the batch size $b$ decreases the number of steps $K$ needed to train a DNN and that there are critical batch sizes minimizing the SFO complexities. To determine whether SGD using the Armijo-line-search learning rate performs better than SGD using a constant learning rate (see the discussion in condition (5)), we numerically compared SGD using the Armijo-line-search learning rate with not only SGD using a constant learning rate but also variants of SGD, such as the momentum method, Adam, AdamW, and RMSProp. We found that SGD using the Armijo-line-search learning rate and the critical batch size performs better than other optimizers in the sense of minimizing the number of steps and the SFO complexities needed to train a DNN (Section 4).

## 2 Mathematical Preliminaries

### 2.1 Definitions

Let $\mathbb{N}$ be the set of nonnegative integers, $[n] := \{1, 2, \ldots, n\}$ for $n \geq 1$, and $[0 : n] := \{0, 1, \ldots, n\}$ for $n \geq 0$. Let $\mathbb{R}^d$ be a $d$–dimensional Euclidean space with inner product $\langle \cdot, \cdot \rangle$ inducing the norm $\|\cdot\|$.

Let $f\colon \mathbb{R}^d \to \mathbb{R}$ be continuously differentiable. We denote the gradient of $f$ by $\nabla f\colon \mathbb{R}^d \to \mathbb{R}^d$. Let $L > 0$. $f\colon \mathbb{R}^d \to \mathbb{R}$ is said to be $L$–smooth if $\nabla f\colon \mathbb{R}^d \to \mathbb{R}^d$ is $L$–Lipschitz continuous, i.e., for all $\boldsymbol{x}, \boldsymbol{y} \in \mathbb{R}^d$, $\|\nabla f(\boldsymbol{x}) - \nabla f(\boldsymbol{y})\| \le L\|\boldsymbol{x} - \boldsymbol{y}\|$. When $f\colon \mathbb{R}^d \to \mathbb{R}$ is $L$–smooth, the following inequality, called the descent lemma (Beck, 2017, Lemma 5.7), holds: for all $\boldsymbol{x}, \boldsymbol{y} \in \mathbb{R}^d$, $f(\boldsymbol{y}) \le f(\boldsymbol{x}) + \langle \nabla f(\boldsymbol{x}), \boldsymbol{y} - \boldsymbol{x}\rangle + \frac{L}{2}\|\boldsymbol{y} - \boldsymbol{x}\|^2$. Let $f_* \in \mathbb{R}$. $f\colon \mathbb{R}^d \to \mathbb{R}$ is said to be bounded below by $f_*$ if, for all $\boldsymbol{x} \in \mathbb{R}^d$, $f(\boldsymbol{x}) \ge f_*$.

## 2.2 Assumptions and problem

Given a parameter $\boldsymbol{\theta} \in \mathbb{R}^d$ and a data point $z$ in a data domain $Z$, a machine learning model provides a prediction whose quality can be measured by a differentiable nonconvex loss function $f(\boldsymbol{\theta}; z)$. We aim to minimize the empirical average loss defined for all $\boldsymbol{\theta} \in \mathbb{R}^d$ by $f(\boldsymbol{\theta}) = \frac{1}{n}\sum_{i\in[n]} f(\boldsymbol{\theta}; z_i) = \frac{1}{n}\sum_{i\in[n]} f_i(\boldsymbol{\theta})$, where $S = (z_1, z_2, \ldots, z_n)$ denotes the training set and $f_i(\cdot) := f(\cdot; z_i)$ denotes the loss function corresponding to the $i$-th training data $z_i$.

This paper considers the following smooth nonconvex optimization problem.

**Problem 2.1** *Suppose that $f_i\colon \mathbb{R}^d \to \mathbb{R}$ ($i \in [n]$) is $L_i$–smooth and bounded below by $f_{i,*}$. Then,*

$$minimize \ f(\boldsymbol{\theta}) := \frac{1}{n}\sum_{i\in[n]} f_i(\boldsymbol{\theta}) \ subject \ to \ \boldsymbol{\theta} \in \mathbb{R}^d.$$

We assume the existence of SFO such that, for a given $\boldsymbol{\theta} \in \mathbb{R}^d$, it returns a stochastic gradient $\mathsf{G}_\xi(\boldsymbol{\theta})$ of the function $f$, where a random variable $\xi$ is supported on a finite/an infinite set $\Xi$ independently of $\boldsymbol{\theta}$. We make the following standard assumptions.

**Assumption 2.1**

*(A1) Let $(\boldsymbol{\theta}_k)_{k\in\mathbb{N}} \subset \mathbb{R}^d$ be the sequence generated by SGD. For each iteration $k$,*

$$\mathbb{E}_{\xi_k}\left[\mathsf{G}_{\xi_k}(\boldsymbol{\theta}_k)\right] = \nabla f(\boldsymbol{\theta}_k), \tag{6}$$

*where $\xi_0, \xi_1, \ldots$ are independent samples, the random variable $\xi_k$ is independent of $(\boldsymbol{\theta}_l)_{l=0}^k$, and $\mathbb{E}_{\xi_k}[\cdot]$ stands for the expectation with respect to $\xi_k$. There exists a nonnegative constant $\sigma^2$ such that*

$$\mathbb{E}_{\xi_k}\left[\|\mathsf{G}_{\xi_k}(\boldsymbol{\theta}_k) - \nabla f(\boldsymbol{\theta}_k)\|^2\right] \le \sigma^2. \tag{7}$$

*(A2) For each iteration $k$, SGD samples a batch $B_k$ of size $b$ independently of $k$ and estimates the full gradient $\nabla f$ as*

$$\nabla f_{B_k}(\boldsymbol{\theta}_k) := \frac{1}{b}\sum_{i\in[b]} \mathsf{G}_{\xi_{k,i}}(\boldsymbol{\theta}_k) = \frac{1}{b}\sum_{i\in[b]} \nabla f_{\xi_{k,i}}(\boldsymbol{\theta}_k),$$

*where $\xi_{k,i}$ is a random variable generated by the $i$-th sampling in the $k$-th iteration.*

From the independence of $\xi_0, \xi_1, \ldots$, we can define the total expectation $\mathbb{E}$ by $\mathbb{E} = \mathbb{E}_{\xi_0}\mathbb{E}_{\xi_1}\cdots\mathbb{E}_{\xi_k}$.

## 2.3 Stochastic gradient descent using Armijo-line-search learning rate

### 2.3.1 Armijo condition

Suppose that $f\colon \mathbb{R}^d \to \mathbb{R}$ is continuously differentiable. We would like to find a stationary point $\boldsymbol{\theta}^\star \in \mathbb{R}^d$ such that $\nabla f(\boldsymbol{\theta}^\star) = \boldsymbol{0}$ by using an iterative method defined by

$$\boldsymbol{\theta}_{k+1} := \boldsymbol{\theta}_k + \alpha_k \boldsymbol{d}_k, \tag{8}$$

where $\alpha_k > 0$ is the step size (called a learning rate in the machine learning field) and $\boldsymbol{d}_k \in \mathbb{R}^d$ is the search direction. Various methods can be used depending on the search direction $\boldsymbol{d}_k$. For example, the method

(8) with $\boldsymbol{d}_k := -\nabla f(\boldsymbol{\theta}_k)$ is gradient descent, while the method (8) with $\boldsymbol{d}_k := -\nabla f(\boldsymbol{\theta}_k) + \beta_{k-1}\boldsymbol{d}_{k-1}$, where $\beta_k \geq 0$, is the conjugate gradient method. If we define $\boldsymbol{d}_k$ (e.g., $\boldsymbol{d}_k := -\nabla f(\boldsymbol{\theta}_k)$), it is desirable to set $\alpha_k^\star$ satisfying

$$f(\boldsymbol{\theta}_k + \alpha_k^\star \boldsymbol{d}_k) = \min_{\alpha>0} f(\boldsymbol{\theta}_k + \alpha \boldsymbol{d}_k). \tag{9}$$

The step size $\alpha_k^\star$ defined by (9) can be easily computed when $f$ is quadratic and convex. However, for a general nonconvex function $f$, it is difficult to compute the step size $\alpha_k^\star$ in (9) exactly. Here, we can use the *Armijo condition* for finding an appropriate step size $\alpha_k$: Let $c \in (0,1)$. We would like to find $\alpha_k > 0$ such that

$$f(\boldsymbol{\theta}_k + \alpha_k \boldsymbol{d}_k) \leq f(\boldsymbol{\theta}_k) + c\alpha_k \langle \nabla f(\boldsymbol{\theta}_k), \boldsymbol{d}_k \rangle. \tag{10}$$

When $\boldsymbol{d}_k$ satisfies the descent property defined by $\langle \nabla f(\boldsymbol{\theta}_k), \boldsymbol{d}_k \rangle < 0$ (e.g., gradient descent using $\boldsymbol{d}_k := -\nabla f(\boldsymbol{\theta}_k)$ has the property such that $\langle \nabla f(\boldsymbol{\theta}_k), \boldsymbol{d}_k \rangle = -\|\nabla f(\boldsymbol{\theta}_k)\|^2 < 0$), the Armijo condition ensures that $f(\boldsymbol{\theta}_{k+1}) = f(\boldsymbol{\theta}_k + \alpha_k \boldsymbol{d}_k) < f(\boldsymbol{\theta}_k)$. Accordingly, $\alpha_k$ satisfying the Armijo condition (10) is appropriate in the sense of minimizing $f$.

The existence of step sizes satisfying the Armijo condition (10) is guaranteed.

**Proposition 2.1** (Nocedal & Wright, 2006, Lemma 3.1) *Let $f\colon \mathbb{R}^d \to \mathbb{R}$ be continuously differentiable. Let $\boldsymbol{\theta}_k \in \mathbb{R}^d$ and let $\boldsymbol{d}_k \,(\neq \boldsymbol{0})$ have the descent property defined by $\langle \nabla f(\boldsymbol{\theta}_k), \boldsymbol{d}_k \rangle < 0$. Let $c \in (0,1)$. Then, there exists $\gamma_k > 0$ such that, for all $\alpha_k \in (0, \gamma_k]$, the Armijo condition (10) holds.*

### 2.3.2 Stochastic gradient descent under Armijo condition

The objective of this paper is to solve Problem 2.1 using mini-batch SGD under Assumption 2.1 defined by

$$\begin{aligned} \boldsymbol{\theta}_{k+1} = \boldsymbol{\theta}_k + \alpha_k \boldsymbol{d}_k &= \boldsymbol{\theta}_k - \alpha_k \nabla f_{B_k}(\boldsymbol{\theta}_k) \\ &= \boldsymbol{\theta}_k - \frac{\alpha_k}{b} \sum_{i \in [b]} \mathsf{G}_{\xi_{k,i}}(\boldsymbol{\theta}_k), \end{aligned}$$

where $b > 0$ is the batch size and $\alpha_k > 0$ is the learning rate. For each iteration $k$, we can use $\boldsymbol{\theta}_k$, $f_{B_k}$, and $\nabla f_{B_k}$. Hence, the Armijo condition (Vaswani et al., 2019, (1)) written as

$$f_{ik}(\boldsymbol{\theta}_k - \alpha_k \nabla f_{ik}(\boldsymbol{\theta}_k)) \leq f_{ik}(\boldsymbol{\theta}_k) - c\alpha_k \|\nabla f_{ik}(\boldsymbol{\theta}_k)\|^2,$$

where $ik$ is the training sample for iteration $k$, at the $k$-th iteration for SGD can be obtained by replacing $f$ in (10) with $f_{B_k}$ and using $\boldsymbol{d}_k = -\nabla f_{B_k}(\boldsymbol{\theta}_k)$:

$$f_{B_k}(\boldsymbol{\theta}_k - \alpha_k \nabla f_{B_k}(\boldsymbol{\theta}_k)) \leq f_{B_k}(\boldsymbol{\theta}_k) - c\alpha_k \|\nabla f_{B_k}(\boldsymbol{\theta}_k)\|^2. \tag{11}$$

The Armijo condition (11) ensures that $f_{B_k}(\boldsymbol{\theta}_{k+1}) = f_{B_k}(\boldsymbol{\theta}_k - \alpha_k \nabla f_{B_k}(\boldsymbol{\theta}_k)) < f_{B_k}(\boldsymbol{\theta}_k)$; i.e., the Armijo condition (11) is appropriate in the sense of minimizing the estimated objective function $f_{B_k}$ from the full objective function $f$. In fact, the numerical results in (Vaswani et al., 2019) indicate that SGD using the Armijo condition (11) is superior to using other deep-learning optimizers to train DNNs.

Algorithm 1 is the SGD algorithm using the Armijo condition (11).

The search direction of Algorithm 1 is $\boldsymbol{d}_k = -\nabla f_{B_k}(\boldsymbol{\theta}_k) \,(\neq \boldsymbol{0})$ which has the descent property defined by $\langle \nabla f_{B_k}(\boldsymbol{\theta}_k), \boldsymbol{d}_k \rangle = -\|\nabla f_{B_k}(\boldsymbol{\theta}_k)\|^2 < 0$. Hence, from Proposition 2.1, there exists a learning rate $\alpha_k \in (0, \gamma_k]$ satisfying the Armijo condition (11). Moreover, the proposition guarantees that the learning rate can be chosen to be sufficiently small, e.g., $\liminf_{k \to +\infty} \alpha_k = 0$.

The convergence analyses of Algorithm 1 use a lower bound of $\alpha_k \in (0, \gamma_k]$ satisfying the Armijo condition (11). To guarantee the existence of such a lower bound, we use the backtracking method ((Nocedal & Wright, 2006, Algorithm 3.1) and (Vaswani et al., 2019, Algorithm 2)) described in Algorithm 1.

The following lemma guarantees the existence of a lower bound on the learning rates computed by Algorithm 1. The proof is given in Appendix A.1.

---

**Algorithm 1** Stochastic gradient descent using Armijo-line-search learning rate

---

**Require:** $c, \delta \in (0, 1)$ (hyperparameter), $b > 0$ (batch size), $\boldsymbol{\theta}_0 \in \mathbb{R}^d$ (initial point), $K \geq 1$ (steps), $\alpha$ (Initialization, see Algorithm 2), $\boldsymbol{\theta}_k \in \mathbb{R}^d$, $f_{B_k} \colon \mathbb{R}^d \to \mathbb{R}$
**Ensure:** $\boldsymbol{\theta}_K \in \mathbb{R}^d$
  $k \leftarrow 0$
  **for** $k = 0, 1, \ldots, K - 1$ **do**
    **while** $\alpha$ satisfying $f_{B_k}(\boldsymbol{\theta}_k - \alpha \nabla f_{B_k}(\boldsymbol{\theta}_k)) > f_{B_k}(\boldsymbol{\theta}_k) - c\alpha \|\nabla f_{B_k}(\boldsymbol{\theta}_k)\|^2$ **do**
      $\alpha \leftarrow \delta\alpha$
    **end while**
    $\alpha_k \leftarrow \alpha$
    Compute $\boldsymbol{\theta}_{k+1} = \boldsymbol{\theta}_k - \alpha_k \nabla f_{B_k}(\boldsymbol{\theta}_k)$
  **end for**

---

**Lemma 2.1** *Consider Algorithm 1 under Assumption 2.1 for solving Problem 2.1. Let $\alpha_k$ be a learning rate satisfying the Armijo condition (11) (whose existence is guaranteed by Proposition 2.1), let $L_{B_k}$ be the Lipschitz constant of $\nabla f_{B_k}$. Then, the following hold.*

(i) [Counter-example of (Vaswani et al., 2019, Lemma 1)] *There exists Problem 2.1 such that $\alpha_k$ does not satisfy $\min\{\frac{2(1-c)}{L_{B_k}}, \overline{\alpha}\} \leq \alpha_k$, where $\overline{\alpha}$ is an upper bound of $\alpha_k$.*

(ii) [Lower bound on learning rate determined by backtracking line search method] *If $\alpha_k$ can be computed by Algorithm 1, then there exists a lower bound of $\alpha_k$ such that $0 < \underline{\alpha} := \frac{2\delta(1-c)}{L} \leq \alpha_k$, where $L$ is the maximum value of $L_i$.*

# 3 Analysis of SGD using Armijo-Line-Search Learning Rate

## 3.1 Convergence analysis of Algorithm 1

Here, we present a convergence analysis of Algorithm 1. The proof of Theorem 3.1 is given in Appendix A.2.

**Theorem 3.1 (Upper bound of the squared norm of the full gradient)** *Consider the sequence $(\boldsymbol{\theta}_k)_{k \in \mathbb{N}}$ generated by Algorithm 1 under Assumption 2.1 for solving Problem 2.1 and suppose that the learning rate $\alpha_k \in [\underline{\alpha}, \overline{\alpha}]$ is computed by Algorithm 1. Then, for all $K \geq 1$, the following hold:*

(i) *In the case of $\frac{1}{L_n} \geq \overline{\alpha}$,*

$$\min_{k \in [0:K-1]} \mathbb{E}\left[\|\nabla f(\boldsymbol{\theta}_k)\|^2\right] \leq \underbrace{\overbrace{\frac{2(f(\boldsymbol{\theta}_0) - f_*)}{(2 - L_n\overline{\alpha})\underline{\alpha}}}^{C_1} \frac{1}{K}}_{B(\boldsymbol{\theta}_0, K)} + \underbrace{\overbrace{\frac{\{(L_n\overline{\alpha} - 1)\underline{\alpha} + \overline{\alpha}\}\sigma^2}{(2 - L_n\overline{\alpha})\underline{\alpha}}}^{C_2} \frac{1}{b}}_{V(\sigma^2, b)},$$

*where $L_n := \frac{1}{n}\sum_{i\in[n]} L_i$, $f_* := \frac{1}{n}\sum_{i\in[n]} f_{i,*}$, $\underline{\alpha} := \frac{2\delta(1-c)}{L}$, $\delta \in (0, 1)$, and $c \in (0, 1)$.*

(ii) *Suppose that the random variable $\xi_k$ follows a discrete uniform distribution $\mathrm{DU}_b(n)$. In the case of $\frac{1}{L_n} < \overline{\alpha} < \hat{\alpha} < \frac{2}{L_n}$,*

$$\min_{k \in [0:K-1]} \mathbb{E}\left[\|\nabla f(\boldsymbol{\theta}_k)\|^2\right] \leq \underbrace{\overbrace{\frac{2(f(\boldsymbol{\theta}_0) - f_*)}{\tilde{\alpha} - (L_n\overline{\alpha} - 1)\overline{\alpha}}}^{C_1} \frac{1}{K}}_{B(\boldsymbol{\theta}_0, K)} + \underbrace{\overbrace{\frac{L_n\overline{\alpha}^2\sigma^2}{\tilde{\alpha} - (L_n\overline{\alpha} - 1)\overline{\alpha}}}^{C_2} \frac{1}{b}}_{V(\sigma^2, b)},$$

*where $L_n := \frac{1}{n}\sum_{i\in[n]} L_i$, $f_* := \frac{1}{n}\sum_{i\in[n]} f_{i,*}$, $\tilde{\alpha} := \frac{2\delta(1-c)}{L_n}$, $\hat{\alpha} := \frac{1+\sqrt{1+8\delta(1-c)}}{2L_n}$, $\delta \in (\frac{1}{4}, 1)$, and $c \in (0, 1 - \frac{1}{4\delta})$.*

Here, we sketch a proof of Theorem 3.1 (A detailed proof of Theorem 3.1 is given in Appendix A.2).

Proof outline of Theorem 3.1(i):

i. We show that there exists a lower bound $\underline{\alpha}$ of $\alpha_k$ satisfying the Armijo condition such that $0 < \underline{\alpha} := \frac{2\delta(1-c)}{L} \leq \alpha_k$ (Lemma 2.1(ii)).

ii. The descent lemma leads to the finding that

$$f(\boldsymbol{\theta}_{k+1}) \leq f(\boldsymbol{\theta}_k) - \frac{\alpha_k}{2}\|\nabla f(\boldsymbol{\theta}_k)\|^2 + \frac{1}{2}(L_n\alpha_k - 1)\alpha_k\|\nabla f_{B_k}(\boldsymbol{\theta}_k)\|^2 + \frac{\alpha_k}{2}\|\nabla f(\boldsymbol{\theta}_k) - \nabla f_{B_k}(\boldsymbol{\theta}_k)\|^2.$$

iii. Under (A1) and (A2), mini-batch stochastic gradient $\nabla f_{B_k}(\boldsymbol{\theta}_k)$ satisfies the equation $\mathbb{E}_{\xi_k}[\nabla f_{B_k}(\boldsymbol{\theta}_k)|\boldsymbol{\theta}_k] = \nabla f(\boldsymbol{\theta}_k)$, $\mathbb{E}_{\xi_k}[\|\nabla f_{B_k}(\boldsymbol{\theta}_k) - \nabla f(\boldsymbol{\theta}_k)\|^2|\boldsymbol{\theta}_k] \leq \frac{\sigma^2}{b}$ (see (24)). Then, we have $\mathbb{E}_{\xi_k}[\|\nabla f_{B_k}(\boldsymbol{\theta}_k)\|^2|\boldsymbol{\theta}_k] \leq \|\nabla f(\boldsymbol{\theta}_k)\|^2 + \frac{\sigma^2}{b}$ (see (25)).

iv. Items i, ii, and iii above, together with the conditions $0 < \underline{\alpha} \leq \alpha_k \leq \overline{\alpha}$ and $\frac{1}{L_n} \geq \overline{\alpha}$, lead to

$$\mathbb{E}_{\xi_k}[f(\boldsymbol{\theta}_{k+1})|\boldsymbol{\theta}_k] \leq f(\boldsymbol{\theta}_k) - \frac{\underline{\alpha}}{2}\|\nabla f(\boldsymbol{\theta}_k)\|^2 + \left\{\frac{(L_n\overline{\alpha} - 1)\underline{\alpha}}{2}\right\}\|\nabla f(\boldsymbol{\theta}_k)\|^2 + \frac{\{(L_n\overline{\alpha} - 1)\underline{\alpha} + \overline{\alpha}\}\sigma^2}{2b},$$

which implies that, for all $k \in \mathbb{N}$,

$$\frac{\underline{\alpha} - (L_n\overline{\alpha} - 1)\underline{\alpha}}{2}\mathbb{E}[\|\nabla f(\boldsymbol{\theta}_k)\|^2] \leq \mathbb{E}[f(\boldsymbol{\theta}_k) - f(\boldsymbol{\theta}_{k+1})] + \frac{\{(L_n\overline{\alpha} - 1)\underline{\alpha} + \overline{\alpha}\}\sigma^2}{2b}.$$

Summing the above inequality from $k = 0$ to $k = K - 1$ ensures that

$$\frac{1}{K}\sum_{k=0}^{K-1}\mathbb{E}\left[\|\nabla f(\boldsymbol{\theta}_k)\|^2\right] \leq \frac{2(f(\boldsymbol{\theta}_0) - f_*)}{\{\underline{\alpha} - (L_n\overline{\alpha} - 1)\underline{\alpha}\}K} + \frac{\{(L_n\overline{\alpha} - 1)\underline{\alpha} + \overline{\alpha}\}\sigma^2}{\{\underline{\alpha} - (L_n\overline{\alpha} - 1)\underline{\alpha}\}b}.$$

Proof outline of Theorem 3.1(ii):

i. Items i, ii, and iii in the proof outline of Theorem 3.1(i), together with the conditions $0 < \underline{\alpha} \leq \alpha_k \leq \overline{\alpha}$ and $\frac{1}{L_n} < \overline{\alpha}$, lead to

$$\mathbb{E}_{\xi_k}[f(\boldsymbol{\theta}_{k+1})] \leq f(\boldsymbol{\theta}_k) - \frac{1}{2}\mathbb{E}_{\xi_k}[\alpha_k]\|\nabla f(\boldsymbol{\theta}_k)\|^2 + \frac{(L_n\overline{\alpha} - 1)\overline{\alpha}}{2}\|\nabla f(\boldsymbol{\theta}_k)\|^2 + \frac{L_n\overline{\alpha}^2\sigma^2}{2b}.$$

ii. Assuming $\xi_k \sim \mathrm{DU}_b(n)$ and applying Jensen's inequality, we have $\mathbb{E}_{\xi_k}[\alpha_k] \geq \frac{2\delta(1-c)}{L_n}$. Hence, we have

$$\frac{\tilde{\alpha} - (L_n\overline{\alpha} - 1)\overline{\alpha}}{2}\sum_{k=0}^{K-1}\mathbb{E}\left[\|\nabla f(\boldsymbol{\theta}_k)\|^2\right] \leq \mathbb{E}[f(\boldsymbol{\theta}_0) - f_*] + \frac{L_n\overline{\alpha}^2\sigma^2 K}{2b},$$

where $\tilde{\alpha} := \frac{2\delta(1-c)}{L_n}$ and $f_* := \frac{1}{n}\sum_{i\in[n]}f_{i,*}$.

iii. Let $\delta \in (\frac{1}{4}, 1)$, $c \in (0, 1 - \frac{1}{4\delta})$, and $\hat{\alpha} := \frac{1 + \sqrt{1 + 8\delta(1-c)}}{2L_n} < \frac{2}{L_n}$. Since $\tilde{\alpha} \leq \overline{\alpha} < \hat{\alpha}$, we have $\tilde{\alpha} - (L_n\overline{\alpha} - 1)\overline{\alpha} > 0$, which implies

$$\frac{1}{K}\sum_{k=0}^{K-1}\mathbb{E}\left[\|\nabla f(\boldsymbol{\theta}_k)\|^2\right] \leq \frac{2(f(\boldsymbol{\theta}_0) - f_*)}{\{\tilde{\alpha} - (L_n\overline{\alpha} - 1)\overline{\alpha}\}K} + \frac{L_n\overline{\alpha}^2\sigma^2}{\{\tilde{\alpha} - (L_n\overline{\alpha} - 1)\overline{\alpha}\}b}.$$

Theorem 3.1 indicates that the upper bound of the minimum value of $\mathbb{E}[\|\nabla f(\boldsymbol{\theta}_k)\|^2]$ consists of a bias term $B(\boldsymbol{\theta}_0, K)$ and variance term $V(\sigma^2, b)$. When the number of steps $K$ is large and the batch size $b$ is large, $B(\boldsymbol{\theta}_0, K)$ and $V(\sigma^2, b)$ become small. Therefore, we need to set $K$ large and $b$ large so that Algorithm 1 will approximate a local minimizer of $f$.

Here, we compare Theorem 3.1 with the convergence analysis of SGD using a constant learning rate. SGD using a constant learning rate $\alpha \in (0, \frac{2}{L_n})$ satisfies

$$\min_{k \in [0:K-1]} \mathbb{E}\left[\|\nabla f(\boldsymbol{\theta}_k)\|^2\right] \leq \frac{2(f(\boldsymbol{\theta}_0) - f_*)}{(2 - L_n\alpha)\alpha} \frac{1}{K} + \frac{L_n\alpha\sigma^2}{2 - L_n\alpha} \frac{1}{b} \tag{12}$$

(The proof of (12) is given in Appendix A.5). We need to set a constant learning rate $\alpha \in (0, \frac{2}{L_n})$ depending on the Lipschitz constant $L_n$ of $\nabla f$. However, since computing $L_n$ is NP-hard (Virmaux & Scaman, 2018), it is difficult to set $\alpha \in (0, \frac{2}{L_n})$. Meanwhile, from Theorem 3.1(ii), we need to set $c, \delta \in (0, 1)$ in Algorithm 1 such that $\delta \in (\frac{1}{4}, 1)$ and $c \in (0, 1 - \frac{1}{4\delta})$ (see Section 4 for the performance of Algorithm 1 using $\delta = 0.9$ and small parameters $c$).

We also compare Theorem 3.1 with Theorem 3 in (Vaswani et al., 2019). Theorem 3 in (Vaswani et al., 2019) indicates that, under a strong growth condition with a constant $\rho$ (i.e., $\mathbb{E}_i[\|\nabla f_i(\boldsymbol{\theta})\|^2] \leq \rho\|\nabla f(\boldsymbol{\theta})\|^2$ ($\boldsymbol{\theta} \in \mathbb{R}^d$)) and the Armijo condition, SGD satisfies that

$$\min_{k \in [0:K-1]} \mathbb{E}\left[\|\nabla f(\boldsymbol{\theta}_k)\|^2\right] \leq \frac{f(\boldsymbol{\theta}_0) - f(\boldsymbol{\theta}^\star)}{\Delta K},$$

where $L$ is the maximum value of the Lipschitz constant $L_i$ of $\nabla f_i$, $c > 1 - \frac{L}{\rho L_n}$, $\overline{\alpha} < \frac{2}{\rho L_n}$, $\Delta := (\overline{\alpha} + \frac{2(1-c)}{L}) - \rho(\overline{\alpha} - \frac{2(1-c)}{L} + L_n\overline{\alpha}^2)$, and $\boldsymbol{\theta}^\star$ is a local minimizer of $f$. Theorem 3.1 is a convergence analysis of Algorithm 1 without assuming the strong growth condition. Moreover, Theorem 3.1 shows that using a large batch size is appropriate for SGD using Armijo line search (Algorithm 1).

### 3.2 Steps needed for $\epsilon$–approximation

To investigate the relationship between the number of steps $K$ needed for nonconvex optimization and the batch size $b$, we consider an $\epsilon$–approximation of Algorithm 1 defined as follows:

$$\min_{k \in [0:K-1]} \mathbb{E}\left[\|\nabla f(\boldsymbol{\theta}_k)\|^2\right] \leq \epsilon^2, \tag{13}$$

where $\epsilon > 0$ is the precision.

Theorem 3.1 leads to the following theorem indicating the relationship between $b$ and the values of $K$ that achieves an $\epsilon$–approximation. The proof of Theorem 3.2 is given in Appendix A.3.

**Theorem 3.2 (Steps needed for nonconvex optimization of SGD using Armijo line search)**
*Suppose that the assumptions in Theorem 3.1 hold. Define $K \colon \mathbb{R} \to \mathbb{R}$ for all $b > \frac{C_2}{\epsilon^2}$ by*

$$K(b) = \frac{C_1 b}{\epsilon^2 b - C_2}, \tag{14}$$

*where the positive constants $C_1$ and $C_2$ are defined as in Theorem 3.1. Then, the following hold:*

(i) *[Steps needed for nonconvex optimization] $K$ defined by (14) achieves an $\epsilon$–approximation (13).*

(ii) *[Properties of the steps] $K$ defined by (14) is monotone decreasing and convex for $b > \frac{C_2}{\epsilon^2}$.*

Theorem 3.2 ensures that the number of steps $K$ needed for SGD using Armijo line search to be an $\epsilon$–approximation is small when the batch size $b$ is large. Therefore, it is useful to set a sufficiently large batch size in the sense of minimizing the steps needed for an $\epsilon$–approximation of SGD using Armijo line search. In other words, Theorem 3.2 implies that a large batch size accelerates DNN training.

### 3.3 Critical batch size minimizing SFO complexity

The following theorem shows the existence of a critical batch size for SGD using Armijo line search. The proof of Theorem 3.3 is given in Appendix A.4.

**Theorem 3.3 (Existence of critical batch size for SGD using Armijo line search)** *Suppose that the assumptions in Theorem 3.1 hold. Define SFO complexity* $N \colon \mathbb{R} \to \mathbb{R}$ *for the number of steps* $K$, *defined by (14), needed for an* $\epsilon$*–approximation (13) and for a batch size* $b > \frac{C_2}{\epsilon^2}$ *by*

$$N(b) = K(b)b = \frac{C_1 b^2}{\epsilon^2 b - C_2},\tag{15}$$

*where the positive constants* $C_1$ *and* $C_2$ *are defined as in Theorem 3.1(ii). Then, the following hold:*

(i) *[SFO complexity]* $N$ *defined by (15) is convex for* $b > \frac{C_2}{\epsilon^2}$.

(ii) *[Critical batch size] There exists a critical batch size*

$$b^\star = \frac{2C_2}{\epsilon^2} = \frac{\sigma^2}{\epsilon^2} \frac{L_n^2 \overline{\alpha}^2}{\{2(1-c)\delta - (L_n \overline{\alpha} - 1)L_n \overline{\alpha}\}}\tag{16}$$

*such that* $b^\star$ *minimizes the SFO complexity (15).*

While large batch sizes require substantial computational resources, the critical batch size that minimizes SFO is small and practical, highlighting the importance of Theorem 3.3.

The previous results in (Shallue et al., 2019; Zhang et al., 2019; Iiduka, 2022) show that, for deep-learning optimizers, there are critical batch sizes at which the SFO complexities are minimized. We are interested in verifying whether a critical batch size exists for SGD using Armijo line search. Theorem 3.3(ii) indicates that the critical batch size can be obtained from the hyperparameters. The next section numerically examines the relationship between the batch size $b$ and the number of steps $K$ needed for nonconvex optimization and the relationship between $b$ and the SFO complexity $N$ to check if there is a critical batch size $b^\star$ minimizing $N$.

### 3.4 Insights into and relationships among Theorems 3.1, 3.2, and 3.3

Theorem 3.1 provides a convergence analysis of SGD using Armijo line search showing that an upper bound of the gradient norm is represented by $\frac{C_1}{K} + \frac{C_2}{b}$ and that the upper bound decreases as the number of steps $K$ and batch size $b$ increase. Therefore, it is desirable for SGD with Armijo line search to use large values of $K$ and $b$. However, Theorem 3.1 does not directly indicate how large $K$ and $b$ should be for training DNNs. Here, we are interested in clarifying the relationship between the $K$ needed to train a DNN and batch size $b$.

To consider the case in which SGD minimizes the full gradient norm of the loss function, we assume that SGD is an $\epsilon$-approximation defined by $\min_{k \in [0:K-1]} \mathbb{E}[\|\nabla f(\boldsymbol{\theta}_k)\|^2] \leq \epsilon^2$; that is, the upper bound of the gradient norm shown in Theorem 3.1 is less than or equal to a certain small positive value $\epsilon^2$, i.e., $\frac{C_1}{K} + \frac{C_2}{b} \leq \epsilon^2$. Then, we have $K \geq \frac{C_1 b}{\epsilon^2 b - C_2}$. This implies that the number of steps needed to obtain an $\epsilon$-approximation of SGD with Armijo line search is $K(b) = \frac{C_1 b}{\epsilon^2 b - C_2}$ depending on batch size $b$.

On the basis of the above discussion, we aim to clarify the relationship between the required $K$ for training a DNN and batch size $b$. Theorem 3.2 elucidates the relationship between the number of steps $K$ and batch size $b$. In particular, Theorem 3.2(ii) indicates that the number of steps needed to train a DNN is monotonically decreasing and convex with respect to batch size. That is, the number of steps $K$ needed for SGD using Armijo line search to be an $\epsilon$-approximation is small when batch size $b$ is large. Therefore, it is useful to set a sufficiently large batch size in the sense of minimizing the steps needed for an $\epsilon$-approximation of SGD using Armijo line search. That is, a large batch size can accelerate DNN training.

One particularly intriguing question is how large the batch size should be. Here we consider SFO complexity to be the stochastic gradient computation cost. Since a DNN model uses $b$ gradients of the loss function,

SFO complexity is $K(b)b$ when the number of steps needed to train the DNN is $K(b)$, which can be obtained from Theorem 3.2. Theorem 3.3 establishes the relationship between batch size $b$ and SFO complexity $K(b)b = \frac{C_1 b^2}{\epsilon^2 b - C_2}$. In particular, Theorem 3.3(ii) indicates that there is a critical batch size that minimizes SFO complexity $N(b) = K(b)b$, which is a convex function of batch size $b$. Theorem 3.2 demonstrates that, as the batch size increases, the number of required steps decreases, suggesting that increasing the batch size accelerates DNN training. However, excessively large batch sizes demand substantial computational resources. Therefore, as Theorem 3.3 indicates, the critical batch size that minimizes SFO complexity is significant and practical because moderate batch sizes are suitable for implementing SGD.

## 4 Numerical Results

We examined whether the numerical results match our theoretical results (Theorems 3.2 and 3.3). We also compared the performance of Algorithm 1 with the performances of other optimizers, such as SGD with a constant learning rate (SGD), a momentum method (Momentum), Adam, AdamW, and RMSProp. The learning rate and hyperparameters of the five optimizers used in each experiment were determined on the basis of a grid search. Python implementations of the methods used in the numerical experiments are available at https://github.com/iiduka-researches/armijo_linesearch.

The metrics were the number of steps $K$ and the SFO complexity $N = Kb$ indicating for different batch sizes $b$, at time $k$, the number of steps $K$ needed for the average gradient norm over the past $k$ steps to be less than $\epsilon = 0.5$. We used Algorithm 1 with the Armijo-line-search learning rate computed by Algorithm 1 with $\delta = 0.9$, $\overline{\alpha} = 10$ (see https://github.com/IssamLaradji/sls for the setting of parameters), and various values of $c$. We reset the step size each epoch by using Algorithm 2 with $J = \lceil \frac{n}{b} \rceil$.

---

**Algorithm 2** Reset step size

**Require:** $\alpha_{\text{init}}$, $J$, $\alpha_{k-1}, \ldots, \alpha_{k-J+1}$
  **if** $k \leq J$ **then**
    $\alpha_k = \alpha_{\text{init}}$
  **else**
    $\alpha_k = \frac{2}{J} \sum_{j=1}^{J+1} \alpha_{k-j}$
  **end if**
  **return** $\alpha_k$

---

We trained ResNet-18 on the CIFAR-10 dataset ($n = 50000$). Figure 1 plots the number of steps needed to train ResNet-18 on the CIFAR-10 dataset for Algorithm 1 versus the batch size. It can be seen that Algorithm 1 reduced the number of steps as the batch size was increased. Figure 2 plots the SFO complexities of Algorithm 1 versus the batch size. It indicates that there are critical batch sizes that minimize the SFO complexities. The Armijo condition (see (11)) implies that, if $c$ is large, then $\alpha_k$ satisfying the Armijo condition is small, which implies that SGD with a small $\alpha_k$ would not work. In fact, as indicated in Figure 1, for training ResNet-18 on the CIFAR-10 dataset, when the batch size exceeded $2^7$, SGD using Armijo line search with $c = 0.1$ required a greater number of steps for convergence than with $c = 0.01, 0.001$ because the learning rate at $c = 0.1$ was much smaller than that at $c = 0.01, 0.001$ (see Appendix A.8). As a result, as shown in Figure 2, SGD using Armijo line search with $c = 0.1$ had a larger SFO complexity than with $c = 0.01, 0.001$. However, the trend in which smaller values of $c$ resulted in fewer steps and reduced SFO complexity was not observed when training ResNet-18 on the MINIST dataset (Figures 5 and 6). This is because the learning rate increased with the batch size for the CIFAR-10 dataset (see Appendix A.8). Therefore, we can conclude that the appropriate value for hyperparameter $c$ depends on the specific training dataset.

Figures 3 and 4 compare the performance of Algorithm 1 with $c = 0.01$ with those of SGD variants. The figures indicate that, when the batch sizes are from $2^5$ to $2^{11}$, SGD+Armijo (Algorithm 1) performs better than the other optimizers. In particular, the SFO complexity of SGD+Armijo (Algorithm 1) using $c = 0.01$ and the critical batch size ($b^\star = 2^8$) is the smallest of the optimizers for any batch size.

Therefore, we can conclude that Algorithm 1 using the critical batch size $b^\star (= 2^8)$ performs better than other optimizers using any batch size in the sense of minimizing the SFO complexities needed to train ResNet18 on the CIFAR-10 dataset.

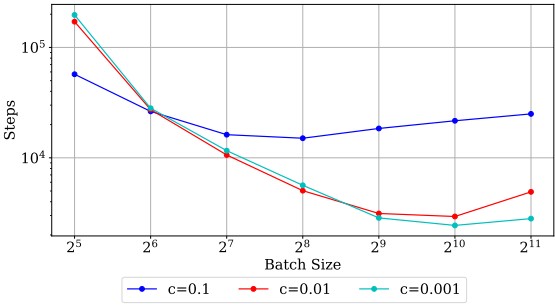

Figure 1: Number of steps for Algorithm 1 versus batch size needed to train ResNet-18 on CIFAR-10

Figure 2: SFO complexity for Algorithm 1 versus batch size needed to train ResNet-18 on CIFAR-10 (The double-circle symbol denotes the measured critical batch size)

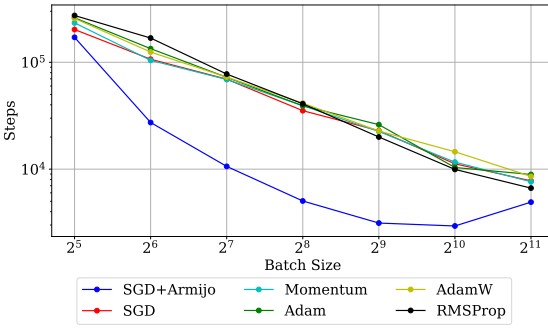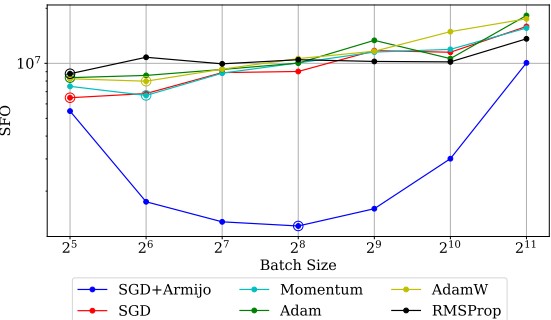

Figure 3: Number of steps for Algorithm 1 with $c = 0.01$ and SGD variants versus batch size needed to train ResNet-18 on CIFAR-10

Figure 4: SFO complexity for Algorithm 1 with $c = 0.01$ and SGD variants versus batch size needed to train ResNet-18 on CIFAR-10 (The double-circle symbol denotes the measured critical batch size)

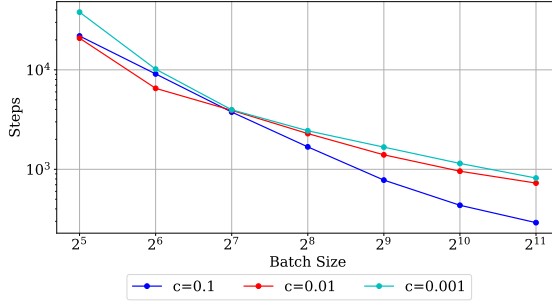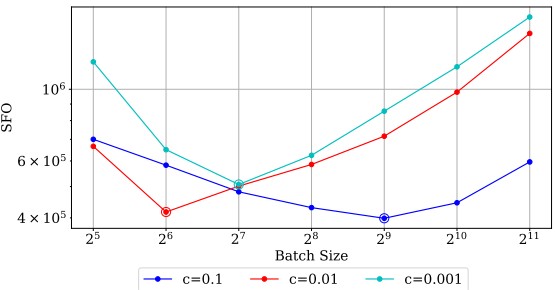

Figure 5: Number of steps for Algorithm 1 versus batch size needed to train ResNet-18 on MNIST

Figure 6: SFO complexity for Algorithm 1 versus batch size needed to train ResNet-18 on MNIST (The double-circle symbol denotes the measured critical batch size)

We also considered the case of training ResNet-18 on the MNIST dataset ($n = 60000$). Figure 5 plots the number of steps needed to train ResNet-18 on the MNIST dataset for Algorithm 1 versus the batch size, and Figure 6 plots the SFO complexities of Algorithm 1 versus the batch size. Here, there are critical batch sizes that minimize the SFO complexities. As in Figures 5 and 6, these figures show that Algorithm 1 decreases the number of steps as the batch size increases and there are critical batch sizes that minimize the SFO complexities.

Figures 7 and 8 compare the performance of Algorithm 1 with $c = 0.1$ with those of SGD variants. Figures 7 and 8 show that the adaptive methods, i.e., Adam and AdamW, performed well. When the batch size was from $2^5$ to $2^8$, SGD performed better than SGD using the Armijo-line-search learning rate in the sense of minimizing the number of steps and SFO complexity. When the batch size was large such as $b = 2^9, 2^{10}$, or $2^{11}$, SGD using the Armijo-line-search learning rate performed better than SGD and Momentum in the sense of minimizing the number of steps and SFO complexity. In short, Figures 3, 4, 7, and 8 show that, when using a large batch size, SGD with Armijo line search performed better than SGD and Momentum.

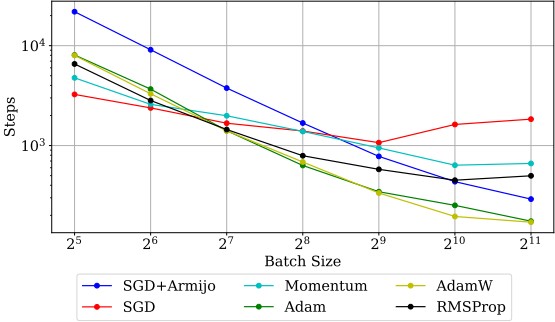
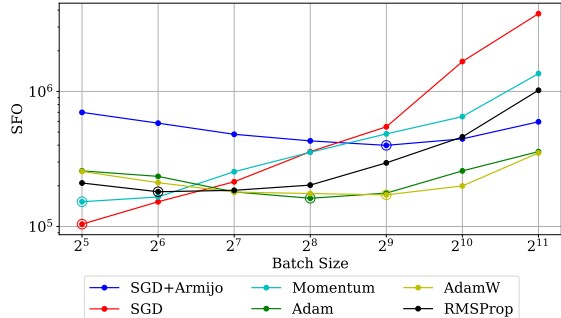

Figure 7: Number of steps for Algorithm 1 with $c = 0.1$ and SGD variants versus batch size needed to train ResNet-18 on MNIST

Figure 8: SFO complexity for Algorithm 1 with $c = 0.1$ and SGD variants versus batch size needed to train ResNet-18 on MNIST (The double-circle symbol denotes the measured critical batch size)

## 5 Conclusion

This paper presented a convergence analysis of SGD using Armijo line search for nonconvex optimization. We showed that the number of steps needed for nonconvex optimization is monotone decreasing and convex with respect to the batch size; i.e., the steps decrease in number as the batch size increases. We also showed that the SFO complexity needed for nonconvex optimization is convex with respect to the batch size and that there exists a critical batch size at which the SFO complexity is minimized. We also presented numerical results that corroborate our theoretical findings. In particular, the numerical results indicated that SGD using Armijo line search and the critical batch size performs better than other optimizers using any batch size in the sense of minimizing the SFO complexity needed to train ResNet-18 on the CIFAR-10 dataset.

## Acknowledgments

We are sincerely grateful to the Action Editor, Alec Koppel, and the three anonymous reviewers for helping us improve the original manuscript. This research is partly supported by the computational resources of the DGX A100 named TAIHO at Meiji University. This work was supported by the Japan Society for the Promotion of Science (JSPS) KAKENHI Grant Number 24K14846 awarded to Hideaki Iiduka.

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

# A  Appendix

## A.1  Proof of Lemma 2.1

(i) Let $k \in \mathbb{N}$ and let $L_{B_k}$ be the Lipschitz constant of $\nabla f_{B_k}$. Lemma 1 in (Vaswani et al., 2019) is as follows:

$$\forall f_{B_k} \colon \mathbb{R}^d \to \mathbb{R} \ \forall c \in (0,1) \ \forall \boldsymbol{\theta}_k \in \mathbb{R}^d \ \forall \overline{\alpha} > 0$$

$$\exists \alpha_k \in (0, \overline{\alpha}] \ (f_{B_k}(\boldsymbol{\theta}_k - \alpha_k \nabla f_{B_k}(\boldsymbol{\theta}_k)) \le f_{B_k}(\boldsymbol{\theta}_k) - c\alpha_k \|\nabla f_{B_k}(\boldsymbol{\theta}_k)\|^2) \Rightarrow \min\left\{\frac{2(1-c)}{L_{B_k}}, \overline{\alpha}\right\} \le \alpha_k. \tag{17}$$

The negative proposition of (17) is as follows:

$$\exists f_{B_k} \colon \mathbb{R}^d \to \mathbb{R} \ \exists c \in (0,1) \ \exists \boldsymbol{\theta}_k \in \mathbb{R}^d \ \exists \overline{\alpha} > 0$$

$$\exists \alpha_k \in (0, \overline{\alpha}] \ (f_{B_k}(\boldsymbol{\theta}_k - \alpha_k \nabla f_{B_k}(\boldsymbol{\theta}_k)) \le f_{B_k}(\boldsymbol{\theta}_k) - c\alpha_k \|\nabla f_{B_k}(\boldsymbol{\theta}_k)\|^2) \wedge \min\left\{\frac{2(1-c)}{L_{B_k}}, \overline{\alpha}\right\} > \alpha_k. \tag{18}$$

We will prove that (18) holds. Let $n = b = 1$, $d = 1$, $c = 0.1$, $\overline{\alpha} = 1$, and $f(\theta) = f_{B_k}(\theta) = \theta^2$. From $\nabla f(\theta) = 2\theta$, we have that $L_{B_k} = 2$. Since $\theta^* = 0$ is the global minimizer of $f$, we set $\theta_k \in \mathbb{R}$ such that $\theta_k \ne \theta^*$. The Armijo condition in this case is such that $(\theta_k - 2\alpha_k \theta_k)^2 \le \theta_k^2 - c\alpha_k(2\theta_k)^2$, which is equivalent to $\alpha_k \le 1 - c = 0.9$. Hence,

$$\exists \alpha_k \in (0,1] \ (\alpha_k \le 0.9) \wedge (\min\{0.9, 1\} > \alpha_k)$$

$$\Leftrightarrow \exists \alpha_k \in (0, \overline{\alpha}] \ (\alpha_k \le 1 - c) \wedge \left(\min\left\{\frac{2(1-c)}{L_{B_k}}, \overline{\alpha}\right\} > \alpha_k\right)$$

$$\Leftrightarrow \exists \alpha_k \in (0, \overline{\alpha}] \ (f_{B_k}(\theta_k - \alpha_k \nabla f_{B_k}(\theta_k)) \le f_{B_k}(\theta_k) - c\alpha_k \|\nabla f_{B_k}(\theta_k)\|^2) \wedge \min\left\{\frac{2(1-c)}{L_{B_k}}, \overline{\alpha}\right\} > \alpha_k,$$

which implies that (18) holds.

(ii) We can show Lemma 2.1(ii) using the proof (Case 2) of Lemma 1 in (Galli et al., 2023). Since $\frac{\alpha_k}{\delta}$ does not satisfy the Armijo condition (11), we have that

$$f_{B_k}\left(\boldsymbol{\theta}_k - \frac{\alpha_k}{\delta}\nabla f_{B_k}(\boldsymbol{\theta}_k)\right) > f_{B_k}(\boldsymbol{\theta}_k) - c\frac{\alpha_k}{\delta}\|\nabla f_{B_k}(\boldsymbol{\theta}_k)\|^2. \tag{19}$$

The $L_{B_k}$–smoothness of $f_{B_k}$ ensures that the descent lemma is true, i.e.,

$$f_{B_k}\left(\boldsymbol{\theta}_k - \frac{\alpha_k}{\delta}\nabla f_{B_k}(\boldsymbol{\theta}_k)\right)$$

$$\le f_{B_k}(\boldsymbol{\theta}_k) + \left\langle \nabla f_{B_k}(\boldsymbol{\theta}_k), \left(\boldsymbol{\theta}_k - \frac{\alpha_k}{\delta}\nabla f_{B_k}(\boldsymbol{\theta}_k)\right) - \boldsymbol{\theta}_k \right\rangle + \frac{L_{B_k}}{2}\left\|\left(\boldsymbol{\theta}_k - \frac{\alpha_k}{\delta}\nabla f_{B_k}(\boldsymbol{\theta}_k)\right) - \boldsymbol{\theta}_k\right\|^2,$$

which implies that

$$f_{B_k}\left(\boldsymbol{\theta}_k - \frac{\alpha_k}{\delta}\nabla f_{B_k}(\boldsymbol{\theta}_k)\right) \le f_{B_k}(\boldsymbol{\theta}_k) + \frac{\alpha_k}{\delta}\left(\frac{L_{B_k}\alpha_k}{2\delta} - 1\right)\|\nabla f_{B_k}(\boldsymbol{\theta}_k)\|^2. \tag{20}$$

Hence, (19) and (20) imply that

$$-c\frac{\alpha_k}{\delta}\|\nabla f_{B_k}(\boldsymbol{\theta}_k)\|^2 \le \frac{\alpha_k}{\delta}\left(\frac{L_{B_k}\alpha_k}{2\delta} - 1\right)\|\nabla f_{B_k}(\boldsymbol{\theta}_k)\|^2,$$

which in turn implies that

$$\frac{\alpha_k}{\delta}\left(\frac{L_{B_k}\alpha_k}{2\delta} - (1-c)\right)\|\nabla f_{B_k}(\boldsymbol{\theta}_k)\|^2 \ge 0.$$

Accordingly,

$$\frac{L_{B_k}\alpha_k}{2\delta} - (1-c) \ge 0, \text{ i.e., } \alpha_k \ge \frac{2\delta(1-c)}{L_{B_k}} \ge \frac{2\delta(1-c)}{L} =: \underline{\alpha}.$$

## A.2 Proof of Theorem 3.1

The definition of $f(\boldsymbol{\theta}) := \frac{1}{n} \sum_{i \in [n]} f_i(\boldsymbol{\theta})$ and the $L_i$–smoothness of $f_i$ $(i \in [n])$ imply that, for all $\boldsymbol{\theta}_1, \boldsymbol{\theta}_2 \in \mathbb{R}^d$,

$$\|\nabla f(\boldsymbol{\theta}_1) - \nabla f(\boldsymbol{\theta}_2)\| \le \frac{1}{n} \sum_{i \in [n]} \|\nabla f_i(\boldsymbol{\theta}_1) - \nabla f_i(\boldsymbol{\theta}_2)\| \le \frac{\sum_{i \in [n]} L_i}{n} \|\boldsymbol{\theta}_1 - \boldsymbol{\theta}_2\|,$$

which in turn implies that $\nabla f$ is Lipschitz continuous with Lipschitz constant $L_n := \frac{1}{n} \sum_{i \in [n]} L_i$. Hence, the descent lemma ensures that, for all $k \in \mathbb{N}$,

$$f(\boldsymbol{\theta}_{k+1}) \le f(\boldsymbol{\theta}_k) + \langle \nabla f(\boldsymbol{\theta}_k), \boldsymbol{\theta}_{k+1} - \boldsymbol{\theta}_k \rangle + \frac{L_n}{2} \|\boldsymbol{\theta}_{k+1} - \boldsymbol{\theta}_k\|^2, \tag{21}$$

which, together with $\boldsymbol{\theta}_{k+1} := \boldsymbol{\theta}_k - \alpha_k \nabla f_{B_k}(\boldsymbol{\theta}_k)$, implies that

$$f(\boldsymbol{\theta}_{k+1}) \le f(\boldsymbol{\theta}_k) - \alpha_k \langle \nabla f(\boldsymbol{\theta}_k), \nabla f_{B_k}(\boldsymbol{\theta}_k) \rangle + \frac{L_n \alpha_k^2}{2} \|\nabla f_{B_k}(\boldsymbol{\theta}_k)\|^2. \tag{22}$$

From $\langle \boldsymbol{x}, \boldsymbol{y} \rangle = \frac{1}{2}(\|\boldsymbol{x}\|^2 + \|\boldsymbol{y}\|^2 - \|\boldsymbol{x} - \boldsymbol{y}\|^2)$ $(\boldsymbol{x}, \boldsymbol{y} \in \mathbb{R}^d)$, we have that, for all $k \in \mathbb{N}$,

$$\langle \nabla f(\boldsymbol{\theta}_k), \nabla f_{B_k}(\boldsymbol{\theta}_k) \rangle = \frac{1}{2} \left( \|\nabla f(\boldsymbol{\theta}_k)\|^2 + \|\nabla f_{B_k}(\boldsymbol{\theta}_k)\|^2 - \|\nabla f(\boldsymbol{\theta}_k) - \nabla f_{B_k}(\boldsymbol{\theta}_k)\|^2 \right).$$

Accordingly, (22) implies that, for all $k \in \mathbb{N}$,

$$f(\boldsymbol{\theta}_{k+1}) \le f(\boldsymbol{\theta}_k) - \frac{\alpha_k}{2} \left( \|\nabla f(\boldsymbol{\theta}_k)\|^2 + \|\nabla f_{B_k}(\boldsymbol{\theta}_k)\|^2 - \|\nabla f(\boldsymbol{\theta}_k) - \nabla f_{B_k}(\boldsymbol{\theta}_k)\|^2 \right) + \frac{L_n \alpha_k^2}{2} \|\nabla f_{B_k}(\boldsymbol{\theta}_k)\|^2$$

$$= f(\boldsymbol{\theta}_k) - \frac{\alpha_k}{2} \|\nabla f(\boldsymbol{\theta}_k)\|^2 + \frac{1}{2}(L_n \alpha_k - 1)\alpha_k \|\nabla f_{B_k}(\boldsymbol{\theta}_k)\|^2 + \frac{\alpha_k}{2} \|\nabla f(\boldsymbol{\theta}_k) - \nabla f_{B_k}(\boldsymbol{\theta}_k)\|^2.$$

(i) We consider the case of $\frac{1}{L_n} \ge \overline{\alpha}$. The condition $0 < \alpha_k \le \overline{\alpha}$ implies that $L_n \alpha_k - 1 \le L_n \overline{\alpha} - 1$ and $0 \ge L_n \overline{\alpha} - 1$. From $0 < \underline{\alpha} = \frac{2\delta(1-c)}{L} \le \alpha_k$, we have that, for all $k \in \mathbb{N}$,

$$f(\boldsymbol{\theta}_{k+1}) \le f(\boldsymbol{\theta}_k) - \frac{\underline{\alpha}}{2} \|\nabla f(\boldsymbol{\theta}_k)\|^2 + \frac{1}{2}(L_n \overline{\alpha} - 1)\underline{\alpha} \|\nabla f_{B_k}(\boldsymbol{\theta}_k)\|^2 + \frac{\overline{\alpha}}{2} \|\nabla f(\boldsymbol{\theta}_k) - \nabla f_{B_k}(\boldsymbol{\theta}_k)\|^2. \tag{23}$$

Assumption 2.1 guarantees that

$$\mathbb{E}_{\xi_k} \left[ \nabla f_{B_k}(\boldsymbol{\theta}_k) | \boldsymbol{\theta}_k \right] = \nabla f(\boldsymbol{\theta}_k) \text{ and } \mathbb{E}_{\xi_k} \left[ \|\nabla f_{B_k}(\boldsymbol{\theta}_k) - \nabla f(\boldsymbol{\theta}_k)\|^2 | \boldsymbol{\theta}_k \right] \le \frac{\sigma^2}{b}. \tag{24}$$

Hence, we have

$$\mathbb{E}_{\xi_k} \left[ \|\nabla f_{B_k}(\boldsymbol{\theta}_k)\|^2 | \boldsymbol{\theta}_k \right]$$
$$= \mathbb{E}_{\xi_k} \left[ \|\nabla f_{B_k}(\boldsymbol{\theta}_k) - \nabla f(\boldsymbol{\theta}_k) + \nabla f(\boldsymbol{\theta}_k)\|^2 | \boldsymbol{\theta}_k \right]$$
$$= \mathbb{E}_{\xi_k} \left[ \|\nabla f_{B_k}(\boldsymbol{\theta}_k) - \nabla f(\boldsymbol{\theta}_k)\|^2 | \boldsymbol{\theta}_k \right] + 2\mathbb{E}_{\xi_k} \left[ \langle \nabla f_{B_k}(\boldsymbol{\theta}_k) - \nabla f(\boldsymbol{\theta}_k), \nabla f(\boldsymbol{\theta}_k) \rangle | \boldsymbol{\theta}_k \right] + \mathbb{E}_{\xi_k} \left[ \|\nabla f(\boldsymbol{\theta}_k)\|^2 | \boldsymbol{\theta}_k \right] \tag{25}$$
$$\le \|\nabla f(\boldsymbol{\theta}_k)\|^2 + \frac{\sigma^2}{b}.$$

Inequalities (23), (24), and (25) guarantee that, for all $k \in \mathbb{N}$,

$$\mathbb{E}_{\xi_k} \left[ f(\boldsymbol{\theta}_{k+1}) | \boldsymbol{\theta}_k \right] \le f(\boldsymbol{\theta}_k) - \frac{\underline{\alpha}}{2} \|\nabla f(\boldsymbol{\theta}_k)\|^2 + \frac{1}{2}(L_n \overline{\alpha} - 1)\underline{\alpha} \left( \|\nabla f(\boldsymbol{\theta}_k)\|^2 + \frac{\sigma^2}{b} \right) + \frac{\overline{\alpha} \sigma^2}{2b}$$

$$= f(\boldsymbol{\theta}_k) - \frac{\underline{\alpha}}{2} \|\nabla f(\boldsymbol{\theta}_k)\|^2 + \left\{ \frac{(L_n \overline{\alpha} - 1)\underline{\alpha}}{2} \right\} \|\nabla f(\boldsymbol{\theta}_k)\|^2 + \frac{\{(L_n \overline{\alpha} - 1)\underline{\alpha} + \overline{\alpha}\} \sigma^2}{2b}. \tag{26}$$

Taking the total expectation on both sides of (26) thus ensures that, for all $k \in \mathbb{N}$,

$$\frac{\underline{\alpha} - (L_n\overline{\alpha} - 1)\underline{\alpha}}{2}\mathbb{E}\left[\|\nabla f(\boldsymbol{\theta}_k)\|^2\right] \leq \mathbb{E}\left[f(\boldsymbol{\theta}_k) - f(\boldsymbol{\theta}_{k+1})\right] + \frac{\{(L_n\overline{\alpha} - 1)\underline{\alpha} + \overline{\alpha}\}\sigma^2}{2b}. \tag{27}$$

Let $K \geq 1$. Summing (27) from $k = 0$ to $k = K - 1$ ensures that

$$\frac{\underline{\alpha} - (L_n\overline{\alpha} - 1)\underline{\alpha}}{2}\sum_{k=0}^{K-1}\mathbb{E}\left[\|\nabla f(\boldsymbol{\theta}_k)\|^2\right] \leq \mathbb{E}\left[f(\boldsymbol{\theta}_0) - f(\boldsymbol{\theta}_K)\right] + \frac{\{(L_n\overline{\alpha} - 1)\underline{\alpha} + \overline{\alpha}\}\sigma^2 K}{2b},$$

which, together with the boundedness of $f$, i.e., $f_* \leq f(\boldsymbol{\theta}_k)$, implies that

$$\frac{\underline{\alpha} - (L_n\overline{\alpha} - 1)\underline{\alpha}}{2}\sum_{k=0}^{K-1}\mathbb{E}\left[\|\nabla f(\boldsymbol{\theta}_k)\|^2\right] \leq \mathbb{E}\left[f(\boldsymbol{\theta}_0) - f_*\right] + \frac{\{(L_n\overline{\alpha} - 1)\underline{\alpha} + \overline{\alpha}\}\sigma^2 K}{2b}.$$

Accordingly,

$$\frac{1}{K}\sum_{k=0}^{K-1}\mathbb{E}\left[\|\nabla f(\boldsymbol{\theta}_k)\|^2\right] \leq \frac{2(f(\boldsymbol{\theta}_0) - f_*)}{\{\underline{\alpha} - (L_n\overline{\alpha} - 1)\underline{\alpha}\}K} + \frac{\{(L_n\overline{\alpha} - 1)\underline{\alpha} + \overline{\alpha}\}\sigma^2}{\{\underline{\alpha} - (L_n\overline{\alpha} - 1)\underline{\alpha}\}b}.$$

Moreover, since we have

$$\min_{k \in [0:K-1]}\mathbb{E}\left[\|\nabla f(\boldsymbol{\theta}_k)\|^2\right] \leq \frac{1}{K}\sum_{k=0}^{K-1}\mathbb{E}\left[\|\nabla f(\boldsymbol{\theta}_k)\|^2\right],$$

the assertion in Theorem 3.1(i) holds.

(ii) Let us consider the case of $\frac{1}{L_n} < \overline{\alpha} < \hat{\alpha} := \frac{1 + \sqrt{1 + 8(1-c)\delta}}{2L_n}$. From $0 < \alpha_k \leq \overline{\alpha}$, we have that, for all $k \in \mathbb{N}$, $L_n\alpha_k - 1 \leq L_n\overline{\alpha} - 1$ and $0 < L_n\overline{\alpha} - 1$. For all $k \in \mathbb{N}$,

$$f(\boldsymbol{\theta}_{k+1}) \leq f(\boldsymbol{\theta}_k) - \frac{\alpha_k}{2}\|\nabla f(\boldsymbol{\theta}_k)\|^2 + \frac{1}{2}(L_n\overline{\alpha} - 1)\overline{\alpha}\|\nabla f_{B_k}(\boldsymbol{\theta}_k)\|^2 + \frac{\overline{\alpha}}{2}\|\nabla f(\boldsymbol{\theta}_k) - \nabla f_{B_k}(\boldsymbol{\theta}_k)\|^2. \tag{28}$$

Inequalities (24), (25), and (28) guarantee that, for all $k \in \mathbb{N}$,

$$\mathbb{E}_{\xi_k}\left[f(\boldsymbol{\theta}_{k+1})|\boldsymbol{\theta}_k\right] \leq f(\boldsymbol{\theta}_k) - \mathbb{E}_{\xi_k}\left[\frac{\alpha_k}{2}\|\nabla f(\boldsymbol{\theta}_k)\|^2\Big|\boldsymbol{\theta}_k\right] + \frac{1}{2}(L_n\overline{\alpha} - 1)\overline{\alpha}\left(\|\nabla f(\boldsymbol{\theta}_k)\|^2 + \frac{\sigma^2}{b}\right) + \frac{\overline{\alpha}\sigma^2}{2b}. \tag{29}$$

Since $\xi_k$ and $\boldsymbol{\theta}_k(\xi_{k-1})$ are independent, we have that

$$\mathbb{E}_{\xi_k}\left[\frac{\alpha_k}{2}\|\nabla f(\boldsymbol{\theta}_k)\|^2\Big|\boldsymbol{\theta}_k\right] = \mathbb{E}_{\xi_k}\left[\frac{\alpha_k}{2}\|\nabla f(\boldsymbol{\theta}_k)\|^2\right] = \frac{1}{2}\|\nabla f(\boldsymbol{\theta}_k)\|^2\mathbb{E}_{\xi_k}\left[\alpha_k\right].$$

Hence, (29) implies that

$$\mathbb{E}_{\xi_k}\left[f(\boldsymbol{\theta}_{k+1})\right] \leq f(\boldsymbol{\theta}_k) - \frac{1}{2}\mathbb{E}_{\xi_k}\left[\alpha_k\right]\|\nabla f(\boldsymbol{\theta}_k)\|^2 + \frac{(L_n\overline{\alpha} - 1)\overline{\alpha}}{2}\|\nabla f(\boldsymbol{\theta}_k)\|^2 + \frac{L_n\overline{\alpha}^2\sigma^2}{2b}. \tag{30}$$

Here, let us assume that $\xi_k \sim \mathrm{DU}_b(n)$. Then, we have that

$$\mathbb{E}_{\xi_k}[L_{B_k}] = \mathbb{E}_{\xi_k}\left[\frac{1}{b}\sum_{i=1}^{b}L_{\xi_{k,i}}\right] = \frac{1}{b}\sum_{i=1}^{b}\mathbb{E}_{\xi_{k,i}}\left[L_{\xi_{k,i}}\right] = \frac{1}{b}\sum_{i=1}^{b}\sum_{j=1}^{n}L_j\mathrm{P}(\xi_{k,i} = j) = \frac{1}{b}\sum_{i=1}^{b}\frac{1}{n}\sum_{j=1}^{n}L_j = L_n.$$

Moreover, Jensen's inequality implies that

$$\overline{\alpha} \geq \mathbb{E}_{\xi_k}\left[\alpha_k\right] \geq \mathbb{E}_{\xi_k}\left[\frac{2\delta(1-c)}{L_{B_k}}\right] \geq \frac{2\delta(1-c)}{\mathbb{E}_{\xi_k}[L_{B_k}]} = \frac{2\delta(1-c)}{L_n} =: \tilde{\alpha}.$$

Hence, (30) ensures that, for all $k \in \mathbb{N}$,

$$\mathbb{E}_{\xi_k} \left[ f(\boldsymbol{\theta}_{k+1}) \right] \leq f(\boldsymbol{\theta}_k) - \frac{1}{2}\tilde{\alpha} \|\nabla f(\boldsymbol{\theta}_k)\|^2 + \frac{(L_n\overline{\alpha} - 1)\overline{\alpha}}{2} \|\nabla f(\boldsymbol{\theta}_k)\|^2 + \frac{L_n\overline{\alpha}^2\sigma^2}{2b}. \tag{31}$$

Taking the total expectation on both sides of (31) thus ensures that, for all $k \in \mathbb{N}$,

$$\frac{\tilde{\alpha} - (L_n\overline{\alpha} - 1)\overline{\alpha}}{2} \mathbb{E}\left[ \|\nabla f(\boldsymbol{\theta}_k)\|^2 \right] \leq \mathbb{E}\left[ f(\boldsymbol{\theta}_k) - f(\boldsymbol{\theta}_{k+1}) \right] + \frac{L_n\overline{\alpha}^2\sigma^2}{2b}. \tag{32}$$

Let $K \geq 1$. Summing (32) from $k = 0$ to $k = K - 1$ ensures that

$$\frac{\tilde{\alpha} - (L_n\overline{\alpha} - 1)\overline{\alpha}}{2} \sum_{k=0}^{K-1} \mathbb{E}\left[ \|\nabla f(\boldsymbol{\theta}_k)\|^2 \right] \leq \mathbb{E}\left[ f(\boldsymbol{\theta}_0) - f(\boldsymbol{\theta}_K) \right] + \frac{L_n\overline{\alpha}^2\sigma^2 K}{2b},$$

which, together with the boundedness of $f$, i.e., $f_* \leq f(\boldsymbol{\theta}_k)$, implies that

$$\frac{\tilde{\alpha} - (L_n\overline{\alpha} - 1)\overline{\alpha}}{2} \sum_{k=0}^{K-1} \mathbb{E}\left[ \|\nabla f(\boldsymbol{\theta}_k)\|^2 \right] \leq \mathbb{E}\left[ f(\boldsymbol{\theta}_0) - f_* \right] + \frac{L_n\overline{\alpha}^2\sigma^2 K}{2b}.$$

Let $\delta \in (\frac{1}{4}, 1)$, $c \in (0, 1 - \frac{1}{4\delta})$, and

$$\hat{\alpha} := \frac{1 + \sqrt{1 + 8(1-c)\delta}}{2L_n} \underset{[1-c<\frac{1}{\delta}]}{<} \frac{2}{L_n}.$$

Then, we have that $4(1-c)\delta > 1$, $1 - \frac{1}{\delta} < 0 < c$, and $1 - c < \frac{1}{\delta}$. Hence,

$$2(1-c)\delta < 2 \Leftrightarrow 2(1-c)\delta - 1 < 1 \Leftrightarrow 8(1-c)\delta \left\{ 2(1-c)\delta - 1 \right\} + 1 < 8(1-c)\delta + 1$$

$$\Leftrightarrow \left\{ 4(1-c)\delta - 1 \right\}^2 < 8(1-c)\delta + 1 \Leftrightarrow 4(1-c)\delta < 1 + \sqrt{8(1-c)\delta + 1}$$

$$\Leftrightarrow \hat{\alpha} := \frac{1 + \sqrt{1 + 8(1-c)\delta}}{2L_n} > \tilde{\alpha} := \frac{2\delta(1-c)}{L_n}.$$

Since $\overline{\alpha} < \hat{\alpha}$, we have that

$$0 < \overline{\alpha} < \hat{\alpha} := \frac{1 + \sqrt{1 + 8(1-c)\delta}}{2L_n} \Leftrightarrow 0 < \overline{\alpha} < \hat{\alpha} = \frac{1 + \sqrt{1 + 4L_n\frac{2(1-c)\delta}{L_n}}}{2L_n}$$

$$\Leftrightarrow 0 < \overline{\alpha} < \hat{\alpha} = \frac{1 + \sqrt{1 + 4L_n\tilde{\alpha}}}{2L_n} \Leftrightarrow \frac{1 - \sqrt{1 + 4L_n\tilde{\alpha}}}{2L_n} < 0 < \overline{\alpha} < \hat{\alpha} = \frac{1 + \sqrt{1 + 4L_n\tilde{\alpha}}}{2L_n}$$

$$\Leftrightarrow -L_n\overline{\alpha}^2 + \overline{\alpha} + \tilde{\alpha} > 0 \Leftrightarrow \tilde{\alpha} - (L_n\overline{\alpha} - 1)\overline{\alpha} > 0.$$

Therefore, we have

$$\frac{1}{K} \sum_{k=0}^{K-1} \mathbb{E}\left[ \|\nabla f(\boldsymbol{\theta}_k)\|^2 \right] \leq \frac{2(f(\boldsymbol{\theta}_0) - f_*)}{\{\tilde{\alpha} - (L_n\overline{\alpha} - 1)\overline{\alpha}\}K} + \frac{L_n\overline{\alpha}^2\sigma^2}{\{\tilde{\alpha} - (L_n\overline{\alpha} - 1)\overline{\alpha}\}b}.$$

## A.3   Proof of Theorem 3.2

(i) We have

$$\frac{C_1}{K} + \frac{C_2}{b} = \epsilon^2$$

is equivalent to

$$K = K(b) = \frac{C_1 b}{\epsilon^2 b - C_2}.$$

Hence, Theorem 3.1 leads to an $\epsilon$–approximation.

(ii) We have

$$\frac{\mathrm{d}K(b)}{\mathrm{d}b} = \frac{-C_1 C_2}{(\epsilon^2 b - C_2)^2} \leq 0 \text{ and } \frac{\mathrm{d}^2 K(b)}{\mathrm{d}b^2} = \frac{2C_1 C_2 \epsilon^2}{(\epsilon^2 b - C_2)^3} \geq 0,$$

which implies that $K$ is monotone decreasing and convex with respect to $b$.

## A.4   Proof of Theorem 3.3

(i) From

$$N(b) = \frac{C_1 b^2}{\epsilon^2 b - C_2},$$

we have

$$\frac{\mathrm{d}N(b)}{\mathrm{d}b} = \frac{C_1 b(\epsilon^2 b - 2C_2)}{(\epsilon^2 b - C_2)^2} \text{ and } \frac{\mathrm{d}^2 N(b)}{\mathrm{d}b^2} = \frac{2C_1 C_2^2}{(\epsilon^2 b - C_2)^3} \geq 0,$$

which implies that $N$ is convex with respect to $b$.

(ii) We have

$$\frac{\mathrm{d}N(b)}{\mathrm{d}b} \begin{cases} < 0 & \text{if } b < b^\star, \\ = 0 & \text{if } b = b^\star = \frac{2C_2}{\epsilon^2}, \\ > 0 & \text{if } b > b^\star. \end{cases}$$

Hence, the point $b^\star$ minimizes $N$.

From Theorem 3.1(ii), we have that

$$L_n := \frac{1}{n} \sum_{i \in [n]} L_i = \frac{2\delta(1-c)}{\tilde{\alpha}}.$$

Hence,

$$\begin{aligned} b^\star &= \frac{2C_2}{\epsilon^2} = \frac{2L_n \overline{\alpha}^2 \sigma^2}{\{\tilde{\alpha} - (L_n \overline{\alpha} - 1)\overline{\alpha}\}\epsilon^2} = \frac{2L_n \overline{\alpha}^2 \sigma^2}{\{(2\delta(1-c)/L_n) - (L_n \overline{\alpha} - 1)\overline{\alpha}\}\epsilon^2} \\ &= \frac{\sigma^2}{\epsilon^2} \frac{L_n^2 \overline{\alpha}^2}{\{2(1-c)\delta - (L_n \overline{\alpha} - 1)L_n \overline{\alpha}\}}. \end{aligned}$$

## A.5   Proof of (12)

Let $K \geq 1$. From (22) and $\alpha_k := \alpha > 0$, we have that, for all $k \in \mathbb{N}$,

$$f(\boldsymbol{\theta}_{k+1}) \leq f(\boldsymbol{\theta}_k) - \alpha\langle \nabla f(\boldsymbol{\theta}_k), \nabla f_{B_k}(\boldsymbol{\theta}_k)\rangle + \frac{L_n \alpha^2}{2}\|\nabla f_{B_k}(\boldsymbol{\theta}_k)\|^2.$$

Hence, (24) and (25) ensure that, for all $k \in \mathbb{N}$,

$$\mathbb{E}\left[f(\boldsymbol{\theta}_{k+1})\right] \leq \mathbb{E}\left[f(\boldsymbol{\theta}_k)\right] - \alpha\mathbb{E}\left[\|\nabla f(\boldsymbol{\theta}_k)\|^2\right] + \frac{L_n \alpha^2}{2}\left(\mathbb{E}\left[\|\nabla f(\boldsymbol{\theta}_k)\|^2\right] + \frac{\sigma^2}{b}\right),$$

which implies that, for all $k \in \mathbb{N}$,

$$\alpha\left(1 - \frac{L_n \alpha}{2}\right)\mathbb{E}\left[\|\nabla f(\boldsymbol{\theta}_k)\|^2\right] \leq \mathbb{E}\left[f(\boldsymbol{\theta}_k) - f(\boldsymbol{\theta}_{k+1})\right] + \frac{L_n \alpha^2 \sigma^2}{2b}.$$

Summing the above inequalities from $k = 0$ to $k = K - 1$ ensures that

$$\alpha\left(1 - \frac{L_n \alpha}{2}\right)\sum_{k=0}^{K-1}\mathbb{E}\left[\|\nabla f(\boldsymbol{\theta}_k)\|^2\right] \leq \mathbb{E}\left[f(\boldsymbol{\theta}_0) - f(\boldsymbol{\theta}_K)\right] + \frac{L_n \alpha^2 \sigma^2 K}{2b}.$$

Since $f$ is bounded below by $f_* := \frac{1}{n}\sum_{i \in [n]} f_{i,*}$, we have

$$\min_{k \in [0:K-1]}\mathbb{E}\left[\|\nabla f(\boldsymbol{\theta}_k)\|^2\right] \leq \frac{1}{K}\sum_{k=0}^{K-1}\mathbb{E}\left[\|\nabla f(\boldsymbol{\theta}_k)\|^2\right] \leq \frac{2\mathbb{E}\left[f(\boldsymbol{\theta}_0) - f_*\right]}{\alpha(2 - L_n \alpha)K} + \frac{L_n \alpha \sigma^2}{(2 - L_n \alpha)b}.$$

### A.6 Estimation of critical batch size

We estimated the critical batch size by using Theorem 3.3(iii) and the ideas presented in (Iiduka, 2022) and (Sato & Iiduka, 2023). We used Algorithm 1 with $c = 0.001$ for training ResNet-18 on the CIFAR-10 dataset (Figure 2). Theorem 3.3(iii) indicates that the equation of the critical batch size involves the unknown value $\sigma^2$. We checked that the Armijo-line-search learning rates for Algorithm 1 with $c = 0.001$ are about 10 (see also (Vaswani et al., 2019, Figure 5 (Left))). Hence, we used $\overline{\alpha} \approx 10$. We estimated the unknown value $X = \frac{\sigma^2}{\epsilon^2}$ in equation (16) of the critical batch size by using $\delta = 0.9$, $b^\star = 2^8$ (see Figure 2) and $\overline{\alpha} \approx 10$ as follows:

$$b^\star = \frac{\sigma^2}{\epsilon^2} \frac{L_n^2 \overline{\alpha}^2}{\{2(1-c)\delta - (L_n\overline{\alpha} - 1)L_n\overline{\alpha}\}}.$$

We consider case of $\frac{1}{L_n} \geq \overline{\alpha}$, since $b^\star$ is monotonically increasing when $L_n \geq 0$. We have

$$b^\star \leq X \frac{1}{2(1-c)\delta}.$$

Setting $c = 0.001$ and $b^\star = 2^8$ (see Figure 2) gives

$$2^8 \geq X \frac{1}{2(1-0.001)0.9}.$$

Let us estimate the critical batch size using $X \approx 460$, and Theorem 3.3(iii). For example, when Algorithm 1 with $c = 0.01$ is used to train ResNet-18 on the CIFAR-10 dataset, the equation of the critical batch size is

$$X \frac{1}{2(1-c)\delta} \approx 258 \approx 2^8 = b^\star,$$

which implies that the estimated critical batch size 258 is close to the measured critical batch size $b^\star = 2^8 = 256$ in Figure 2.

### A.7 Test accuracies of SGD using Armijo-line-search learning rate for training ResNet-18 on CIFAR-10 and MINIST datasets

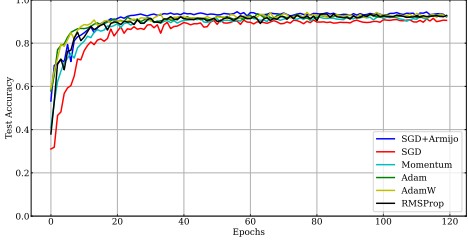
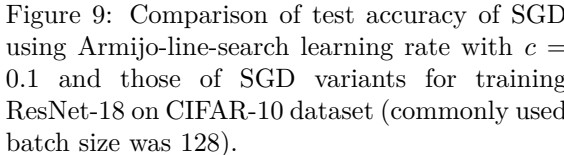
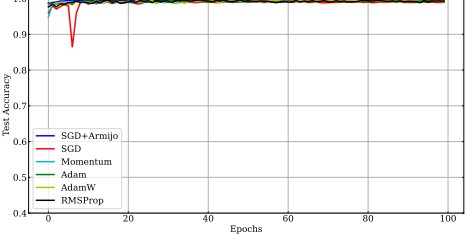

Figure 9: Comparison of test accuracy of SGD using Armijo-line-search learning rate with $c = 0.1$ and those of SGD variants for training ResNet-18 on CIFAR-10 dataset (commonly used batch size was 128).

Figure 10: Comparison of test accuracy of SGD using Armijo-line-search learning rate with $c = 0.1$ and those of SGD variants for training ResNet-18 on MNIST dataset (commonly used batch size was 128).

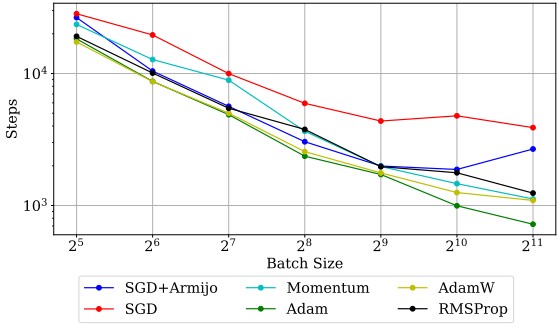

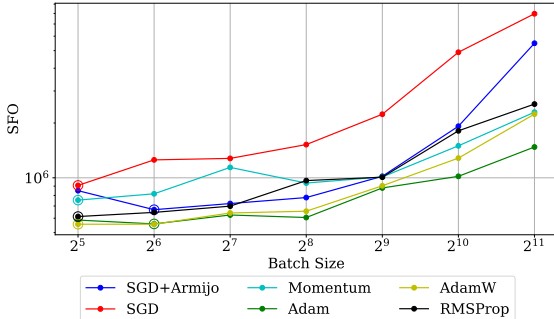

Figure 11: Number of steps for Algorithm 1 with $c = 0.1$ and SGD variants to achieve test accuracy more than 0.90 versus batch size for training ResNet-18 on CIFAR-10 dataset.

Figure 12: SFO complexity for Algorithm 1 with $c = 0.1$ and SGD variants to achieve test accuracy more than 0.90 versus batch size for training ResNet-18 on CIFAR-10 dataset.

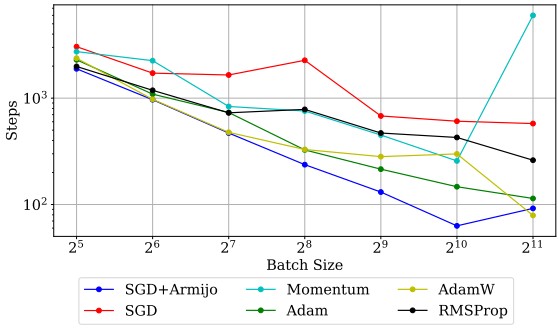

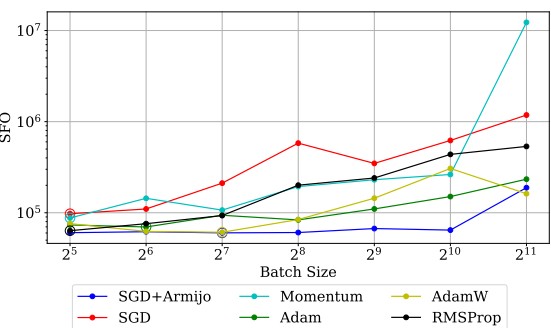

Figure 13: Number of steps for Algorithm 1 with $c = 0.1$ and SGD variants to achieve test accuracy more than 0.99 versus batch size for training ResNet-18 on MNIST dataset.

Figure 14: SFO complexity for Algorithm 1 with $c = 0.1$ and SGD variants to achieve test accuracy more than 0.99 versus batch size for training ResNet-18 on MNIST dataset.

### A.8 Learning rates satisfying Armijo condition for training ResNet-18 on CIFAR-10 and MINIST datasets

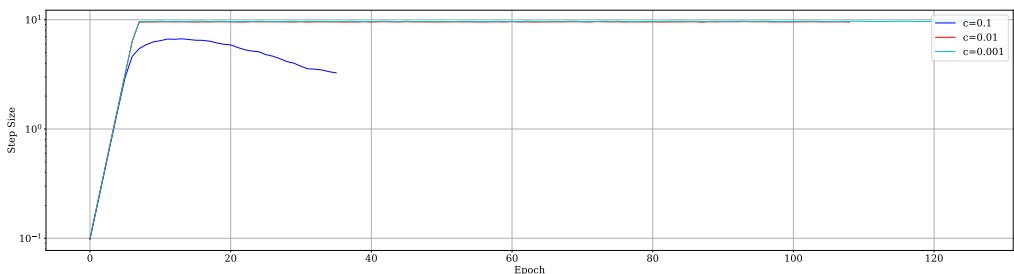

Figure 15: Learning rate adjustment until average gradient norm over previous $k$ steps falls below $\epsilon = 0.5$ when training ResNet-18 on CIFAR-10 dataset with batch size of $2^5$.

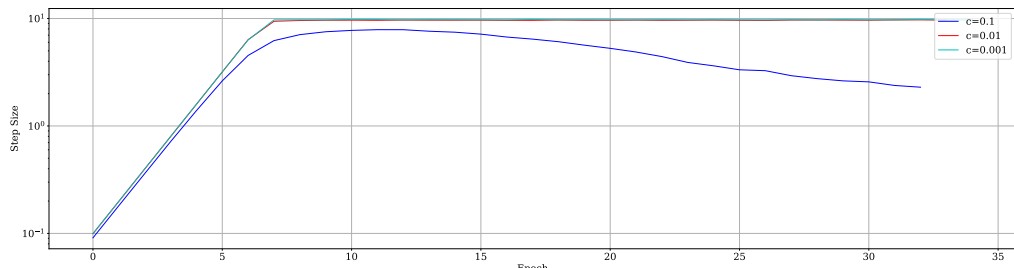

Figure 16: Learning rate adjustment until average gradient norm over previous $k$ steps falls below $\epsilon = 0.5$ when training ResNet-18 on CIFAR-10 dataset with batch size of $2^6$.

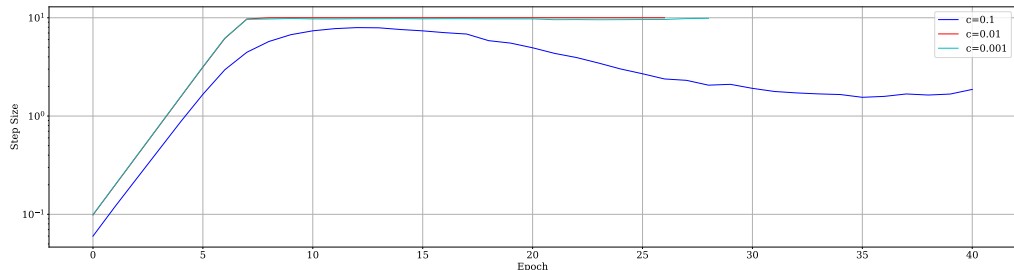

Figure 17: Learning rate adjustment until average gradient norm over previous $k$ steps falls below $\epsilon = 0.5$ when training ResNet-18 on CIFAR-10 dataset with batch size of $2^7$.

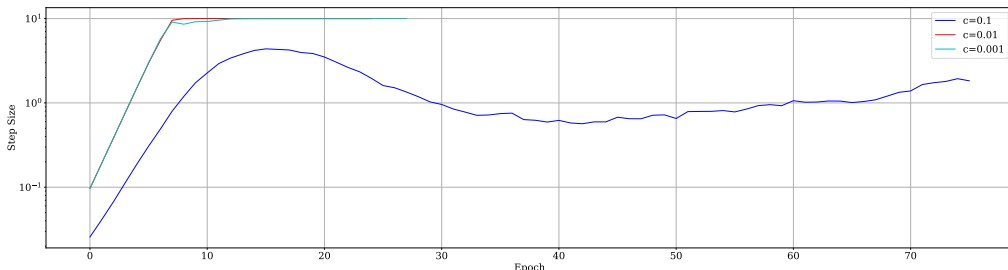

Figure 18: Learning rate adjustment until average gradient norm over previous $k$ steps falls below $\epsilon = 0.5$ when training ResNet-18 on CIFAR-10 dataset with batch size of $2^8$.

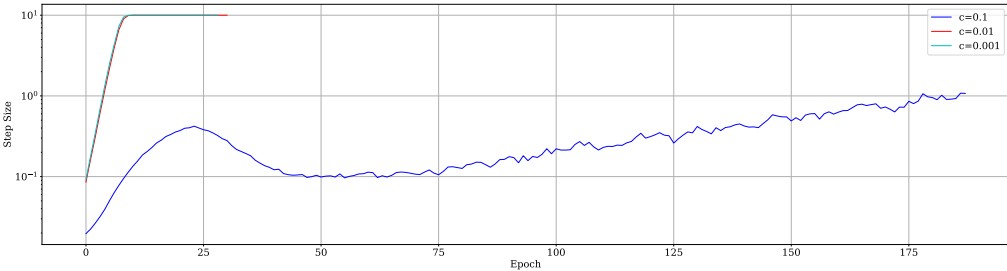

Figure 19: Learning rate adjustment until average gradient norm over previous $k$ steps falls below $\epsilon = 0.5$ when training ResNet-18 on CIFAR-10 dataset with batch size of $2^9$.

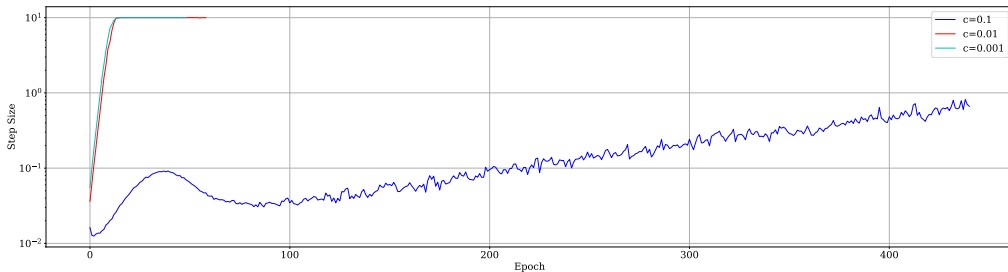

Figure 20: Learning rate adjustment until average gradient norm over previous $k$ steps falls below $\epsilon = 0.5$ when training ResNet-18 on CIFAR-10 dataset with batch size of $2^{10}$.

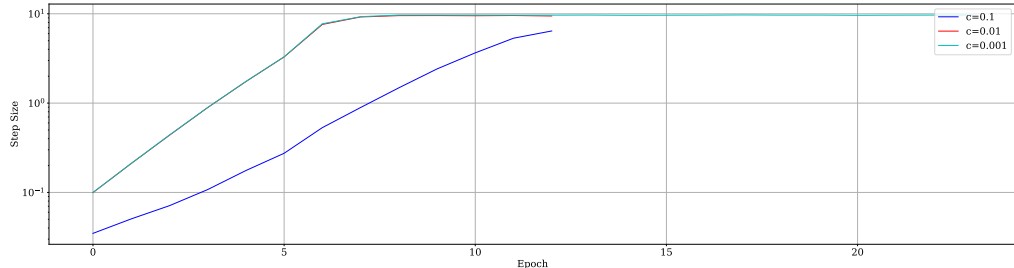

Figure 21: Learning rate adjustment until average gradient norm over previous $k$ steps falls below $\epsilon = 0.5$ when training ResNet-18 on MNIST dataset with batch size of $2^5$.

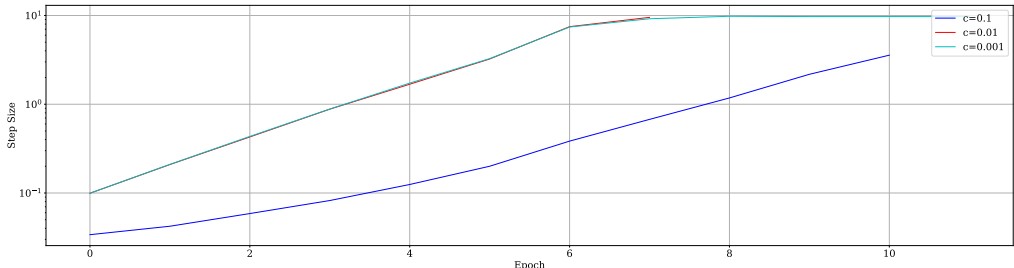

Figure 22: Learning rate adjustment until average gradient norm over previous $k$ steps falls below $\epsilon = 0.5$ when training ResNet-18 on MNIST dataset with batch size of $2^6$.

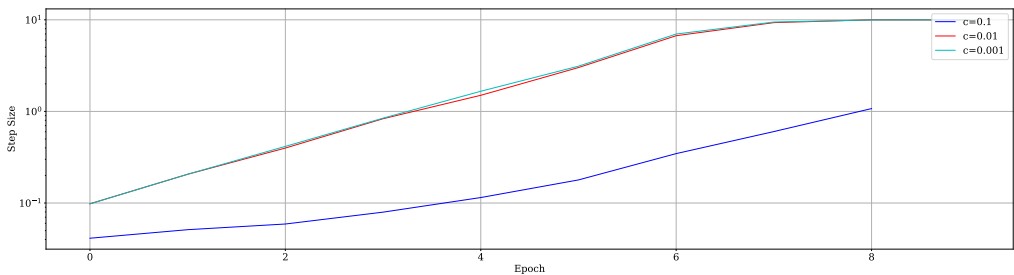

Figure 23: Learning rate adjustment until average gradient norm over previous $k$ steps falls below $\epsilon = 0.5$ when training ResNet-18 on MNIST dataset with batch size of $2^7$.

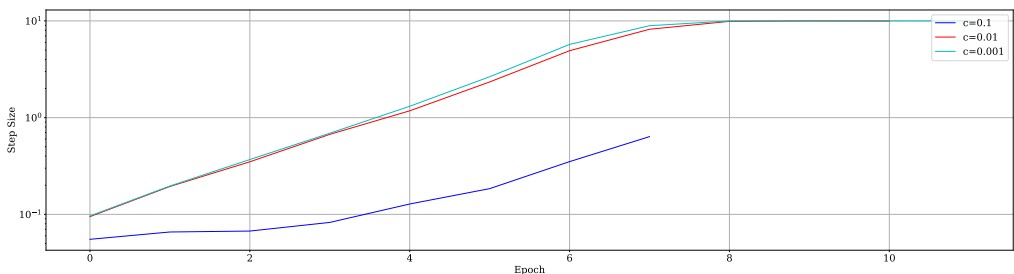

Figure 24: Learning rate adjustment until average gradient norm over previous $k$ steps falls below $\epsilon = 0.5$ when training ResNet-18 on MNIST dataset with batch size of $2^8$.

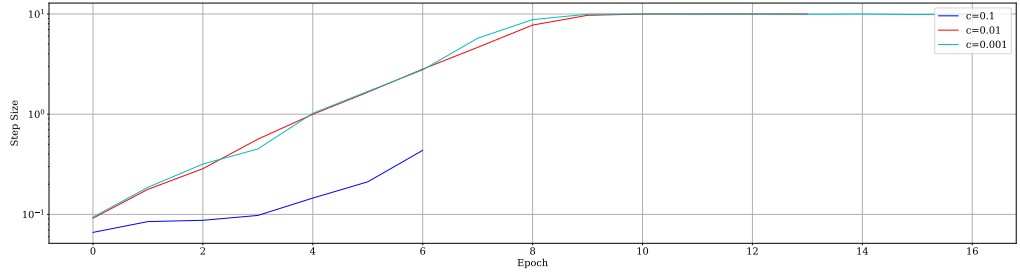

Figure 25: Learning rate adjustment until average gradient norm over previous $k$ steps falls below $\epsilon = 0.5$ when training ResNet-18 on MNIST dataset with batch size of $2^9$.

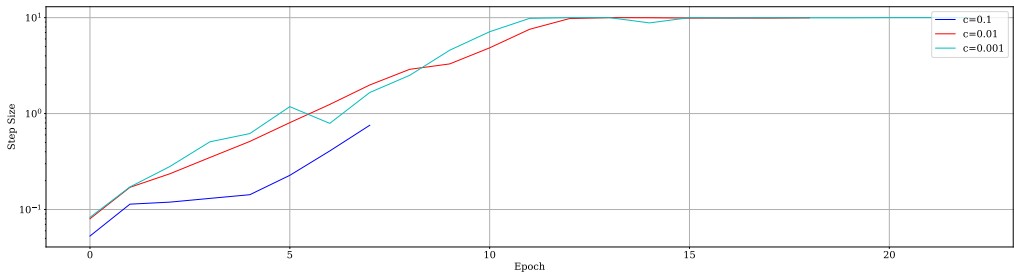

Figure 26: Learning rate adjustment until average gradient norm over previous $k$ steps falls below $\epsilon = 0.5$ when training ResNet-18 on MNIST dataset with batch size of $2^{10}$.

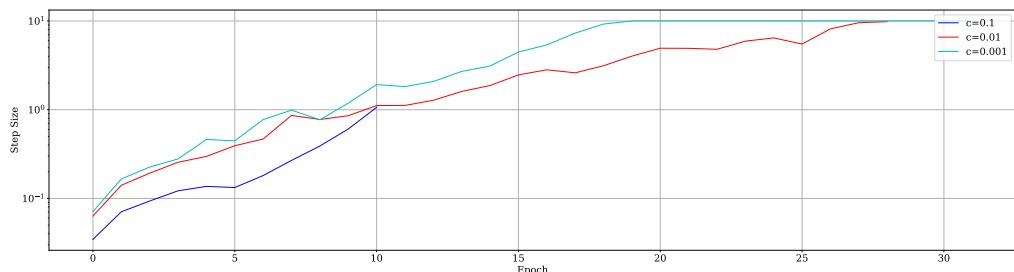

Figure 27: Learning rate adjustment until average gradient norm over previous $k$ steps falls below $\epsilon = 0.5$ when training ResNet-18 on MNIST dataset with batch size of $2^{11}$.

## A.9 Comparisons of SGD using Armijo-line-search learning rate with SGD using constant learning rate $1/\sqrt{K}$ for training ResNet-18 on CIFAR-10 and MINIST datasets

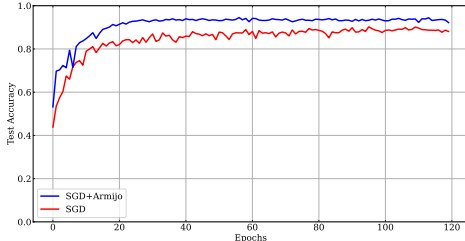 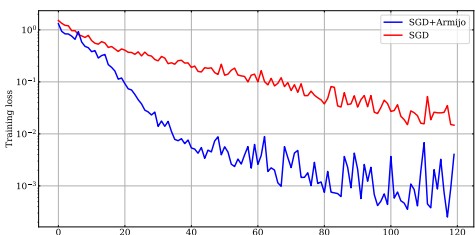

Figure 28: Comparison of test accuracy of SGD using Armijo-line-search learning rate with $c = 0.1$ and that of SGD using constant learning rate $\alpha = 1/\sqrt{K} = 1/\sqrt{(50000/128) \times 120} \approx 0.004$ for training ResNet-18 on CIFAR-10 dataset (commonly used batch size was 128).

Figure 29: Comparison of training loss of SGD using Armijo-line-search learning rate with $c = 0.1$ and that of SGD using constant learning rate $\alpha = 1/\sqrt{K} = 1/\sqrt{(50000/128) \times 120} \approx 0.004$ for training ResNet-18 on CIFAR-10 dataset (commonly used batch size was 128).

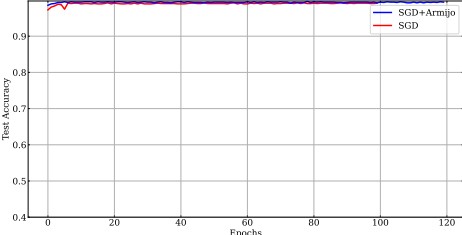

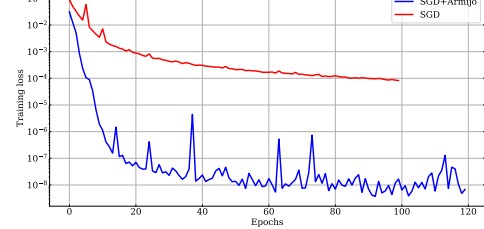

Figure 30: Comparison of test accuracy of SGD using Armijo-line-search learning rate with $c = 0.1$ and that of SGD using constant learning rate $\alpha = 1/\sqrt{K} = 1/\sqrt{(60000/120) \times 100} \approx 0.004$ for training ResNet-18 on MNIST dataset (commonly used batch size was 128).

Figure 31: Comparison of training loss of SGD using Armijo-line-search learning rate with $c = 0.1$ and that of SGD using constant learning rate $\alpha = 1/\sqrt{K} = 1/\sqrt{(60000/128) \times 120} \approx 0.004$ for training ResNet-18 on MNIST dataset (commonly used batch size was 128).

