# OpenReview forum: "Relationship between Batch Size and Number of Steps Needed for Nonconvex Optimization of Stochastic Gradient Descent using Armijo-Line-Search Learning Rate"
_TMLR — Accepted by TMLR_

### Review · Reviewer_UUd5 · 2024-12-11

**Summary Of Contributions:**

This paper presents a convergence analysis of SGD using the Armijo line search for nonconvex optimization. It shows that the number of steps needed for nonconvex optimization is monotonically decreasing and convex with respect to the batch size. It also shows that the SFO complexity needed for nonconvex optimization is convex with respect to the batch size and that there exists a critical batch size at which the SFO complexity is minimized.

**Audience:**

Yes

**Claims And Evidence:**

Yes

**Requested Changes:**

The requested changes about writing is below (I encourage the authors to read through the paper and make changes wherever needed):
1. Abstract: "the previous numerical results showed that SGD ..." -> remove "the"
2. Abstract: "The numerical results indicate that ..." -> This sentence is redundant.
3. Introduction, P1: "Variants have also been presented" -> Variants of what?
4. Introduction, P2: "Not only a batch size but also a learning rate affects the performance of deep-learning" -> change "a" to "the"
5. Introduction, P2: "Here, SGD using a constant learning rate ... satisfies ... " -> change "using" to "uses" and change "satisfies" to "satisfying"
6. Introduction, P2: "Moreover, SGD using a learning rate ... " -> change "using" to "uses"
7. Introduction, P2: "Motivated by the useful numerical results in (Vaswani et al., 2019), we decided to perform convergence analyses ..." and "Hence, in accordance with the first motivation stated above, we decided to investigate appropriate batch sizes for SGD with the Armijo-line-search learning rate. In particular, we were interested in verifying whether a critical ..." -> change the past tense to present
8. Table 1: "Upper Bound C1/K + C2/b" -> "Upper Bound written as C1/K + C2/b", also add equal sign after "C1" and "C2" in the equations
9. Change the past tense for the sentences starting with "we" in Section 1.3.4 to present.
10. Section 2.3.2: "Hence, the Armijo condition (Vaswani et al., 2019, (1)) ..." -> it would be better if the authors could explicitly write what (1) in Vaswani et al., 2019 is for easy reading.
11. Section 2.3.2: "In fact, the numerical results in (Vaswani et al., 2019, Section 7) indicate ..." -> remove "Section 7"

**Strengths And Weaknesses:**

While the theoretical result is no doubt interesting, there are some weaknesses of this paper that need to be addressed. In summary,
1. The writing needs to be improved. (See Requested Changes below.)
2. In the experiments, the authors showed the number of steps needed as a function of batch size. However, in reality, we are more concerned with the metrics related to the actual task we are working on, for example, classification accuracy. It would be interesting if the authors could also present the metric as a function of the number of steps and the batch size.

---

> ### Author Response · Authors · 2025-02-11
> **Replies for Reviewer UUd5's comments**
>
> We appreciate your detailed assessments and helpful feedback. We have revised the manuscript to incorporate all of the recommendations, which has resulted in an improved presentation of our work. The revised parts of the manuscript are marked in red.
>
> **Weakness 1:** The writing needs to be improved. (See Requested Changes below.)
>
> **Reply:** Thank you for pointing this out. We truly appreciate your detailed suggestions. We have revised it based on your suggestions.
>
> **Weakness 2:** In the experiments, the authors showed the number of steps needed as a function of batch size. However, in reality, we are more concerned with the metrics related to the actual task we are working on, for example, classification accuracy. It would be interesting if the authors could also present the metric as a function of the number of steps and the batch size.
>
> **Reply:** Thank you very much for your feedback. We now present the results of additional experiments analyzing the number of steps and batch size with training accuracy as the metric. The updated results are in Appendix A.7 in the revised manuscript.
>
> We also thank you for your interesting comment about the metric depending on both the number of steps $K$ and batch size $b$. The deep neural network model uses $b$ gradients of the loss function per step. Accordingly, the model has a stochastic gradient computation cost of $Kb$, which is called stochastic first-order oracle (SFO) complexity. Therefore, an example of the metric depending on both the number of steps $K$ and batch size $b$ is SFO complexity. We now present numerical results plotting SFO complexity $Kb$ against the batch size $b$ needed to train ResNet-18 on the CIFAR-10 and MNIST datasets in Section 4.
>
> **Requested Changes 1:** Abstract: "the previous numerical results showed that SGD ..." -> remove "the"
>
> **Reply:** Thank you for pointing this out. We have removed "the" from the sentence in the abstract.
>
> **Requested Changes 2:** Abstract: "The numerical results indicate that ..." ->This sentence is redundant.
>
> **Reply:** Thank you for pointing this out. We have removed the redundant sentence from the abstract.
>
>
> **Requested Changes 3:** Introduction, P1: "Variants have also been presented" -> Variants of what?
>
> **Reply:** Thank you for pointing out the lack of clarity. We have specified that it is variants of SGD.
>
>
> **Requested Changes 4:** Introduction, P2: "Not only a batch size but also a learning rate affects the performance of deep-learning" -> change "a" to "the"
>
> **Reply:** Thank you for pointing this out. We have changed "a" to "the".
>
> **Requested Changes 5:** Introduction, P2: "Here, SGD using a constant learning rate ... satisfies ... " -> change "using" to "uses" and change "satisfies" to "satisfying"
>
> **Reply:** Thank you for pointing this out. We have replaced "using" with "uses" and "satisfies" with "satisfying".

---

> ### Author Response · Authors · 2025-02-11
> **Replies for Reviewer UUd5's comments**
>
> **Requested Changes 6:** Introduction, P2: "Moreover, SGD using a learning rate ... " -> change "using" to "uses"
>
> **Reply:** In this case, the main verb in the sentence is “was presented”, so "using" is correct.
>
>
> **Requested Changes 7:** Introduction, P2: "Motivated by the useful numerical results in (Vaswani et al., 2019), we decided to perform convergence analyses ..." and ""Hence, in accordance with the first motivation stated above, we decided to investigate appropriate batch sizes for SGD with the Armijo-line-search learning rate. In particular, we were interested in verifying whether a critical ..." -> change the past tense to present
>
> **Reply:** Our English language consultant pointed out that these sentences are describing decisions we made when we started the work being reported in this paper, so using the present tense may confuse the reader. We have thus left both instances of “decided” in the past tense.
>
> **Requested Changes 8:** Table 1: "Upper Bound C1/K + C2/b" -> "Upper Bound written as C1/K + C2/b", also add equal sign after C1 and C2 in the equations
>
> **Reply:** Thank you for pointing this out. We have updated the description and added the equal signs in the equations as suggested.
>
>
> **Requested Changes 9:** Change the past tense for the sentences starting with "we" in Section 1.3.4 to present.
>
> **Reply:** Our English language consultant pointed out that these sentences are describing things we did during the course of the work being reported in this paper, so using the present tense may confuse the reader. We have thus left these sentences in the past tense.
>
>
> **Requested Changes 10:** Section 2.3.2: "Hence, the Armijo condition (Vaswani et al., 2019, (1)) ..." -> it would be better if the authors could explicitly write what (1) in Vaswani et al., 2019 is for easy reading.
>
> **Reply:** Thank you for pointing this out. We have added an explicit description of (1) from Vaswani et al., 2019, for better clarity.
>
>
> **Requested Changes 11:** Section 2.3.2: "In fact, the numerical results in (Vaswani et al., 2019, Section 7) indicate ..." -> remove "Section 7"
>
> **Reply:** Thank you for pointing this out. We have removed "Section 7".

---

### Review · Reviewer_qD9t · 2024-12-26

**Summary Of Contributions:**

The paper primarily analyzes the convergence of Stochastic Gradient Descent using Armijo Line Search (performed using backtracking) for optimizing non-convex optimization problems with milder smoothness conditions in context of minimizing loss functions of a Neural Network.

The main contribution of the paper is analyzing an expected epsilon critical point in terms of the batch size and number of steps for the nonconvex optimization case with a more general objective function than SoTA.

Additionally, the authors show that the oracle complexity is a convex function under certain conditions and that existence of a critical batch size.

The theoretical results are complemented by some numerical experiments on MNIST and CIFAR-10 dataset used for training Residual Nets.

**Audience:**

Yes

**Claims And Evidence:**

Yes

**Requested Changes:**

1. Suggest reviewing and justifying if the Theorem 3.2 and Theorem 3.3 are indeed worthy of the limelight. TMLR does not focus on novelty as much on the exactness of claims.

2. It would be good to have a proof outline of the main theorem explaining the key steps of the proof in the main text itself. You can use https://arxiv.org/pdf/1308.6594 for inspiration.

3. Clean the insights after the main theorem atleast clearly discussing the various implications of the result.

4. What is $\nabla f_{\xi_k,k}$ in A2?

5. Can Algorithm 1 and Algorithm 2 be combined? I understand Algorithm 2 is way to do Algorithm 1 but is there a cleaner way?

6. In the numerical results a discussion on the comparision with the other methods would be nice. The conclusion "Therefore, we can conclude that Algorithm 1 using the critical batch size b performs as well as other optimizers using any batch size in the sense of minimizing the SFO complexities needed to train ResNet18 on the MNIST dataset." does not seem to have enough justification given the results. Figure 7 and 8 are not even discussed. Each numerical result should ideally have a hypothesis associated with it which it either rejects or fails to.

**Strengths And Weaknesses:**

Strengths:

1. The paper claims are clear with respect to the previous work in the field and it is easy to understand the key assumption being relaxed.

2. The paper has clear writing for the most part.

3. The paper derives the expressions clearly in terms of the parameters of the concern.


Weakness:

1. The authors claim three theoretical results, however I consider Theorem 3.1 to be the main theoretical result and the other results to be straightforward extensions. Would love to discuss if the authors feel otherwise.

2. The proof of the main result is deferred to the appendix, being a theoretical paper there should have been a proof outline in the main text.

3. There are no clear and clean insights for each of the theoretical results.

4. The numerical results are not clearly discussed.

---

> ### Author Response · Authors · 2025-02-11
> **Replies for Reviewer qD9t's comments**
>
> We appreciate your detailed assessments and helpful feedback. We have revised the manuscript to incorporate all of the recommendations, which has resulted in an improved presentation of our work. The revised parts of the manuscript are marked in red.
>
> **Weaknesses 1 and 3:** The authors claim three theoretical results, however I consider Theorem 3.1 to be the main theoretical result and the other results to be straightforward extensions. Would love to discuss if the authors feel otherwise.
>
> There are no clear and clean insights for each of the theoretical results.
>
> **Requested Changes 1 and 3:** Suggest reviewing and justifying if the Theorem 3.2 and Theorem 3.3 are indeed worthy of the limelight. TMLR does not focus on novelty as much on the exactness of claims.
>
> Clean the insights after the main theorem at least clearly discussing the various implications of the result.
>
>
> **Reply:** Thank you very much for your feedback. Theorem 3.1 provides a convergence analysis of SGD with Armijo line search showing that an upper bound of the gradient norm is represented by $\frac{C_1}{K} + \frac{C_2}{b}$ and that the upper bound decreases as the number of steps $K$ and batch size $b$ increase. Hence, it is desirable for SGD with Armijo line search to use large values of $K$ and $b$. However, Theorem 3.1 does not provide direct guidance on the appropriate values for $K$ and $b$ in training DNNs. We are interested in clarifying the relationship between the $K$ needed to train a DNN and batch size $b$.
>
> To consider the case in which SGD minimizes the full gradient norm of the loss function, we assume that SGD is an $\epsilon$-approximation defined by $\min\_{k \in [0:K-1]} \mathbb{E}[\|\nabla f ({\theta}_k)\|^2] \leq \epsilon^2$; that is, the upper bound of the gradient norm shown in Theorem 3.1 is less than or equal to a certain small positive value $\epsilon^2$, i.e.,
> \begin{align*}
> \frac{C_1}{K} + \frac{C_2}{b}
> \leq \epsilon^2.
> \end{align*}
> Then, we have
> \begin{align*}
>  K \geq \frac{C_1 b}{\epsilon^2 b - C_2}.
> \end{align*}
> This implies that the number of steps needed to obtain an $\epsilon$-approximation of SGD using Armijo line search is $K(b) = \frac{C_1 b}{\epsilon^2 b - C_2}$ depending on batch size $b$.
>
> On the basis of the above discussion, we aim to clarify the relationship between the required $K$ for training a DNN and batch size $b$. Theorem 3.2 elucidates the relationship between the number of steps $K$ and batch size $b$. In particular, Theorem 3.2(ii) indicates that the number of steps needed to train a DNN is monotonically decreasing and convex with respect to batch size. That is, the number of steps $K$ needed for SGD using Armijo line search to be an $\epsilon$--approximation is small when batch size $b$ is large. Therefore, it is useful to set a sufficiently large batch size in the sense of minimizing the steps needed for an $\epsilon$--approximation of SGD using Armijo line search. That is, a large batch size accelerates DNN training.
>
> One particularly intriguing question is how large the batch size should be. We consider SFO complexity to be the stochastic gradient computation cost. Since a DNN model uses $b$ gradients of the loss function, SFO complexity is $K(b) b$ when the number of steps needed to train the DNN is $K(b)$, which can be obtained from Theorem 3.2. Theorem 3.3 establishes the relationship between batch size $b$ and SFO complexity $K(b) b = \frac{C_1 b^2}{\epsilon^2 b - C_2}$. In particular, Theorem 3.3(ii) indicates that there is a critical batch size that minimizes SFO complexity $N(b) = K(b) b$, which is a convex function of batch size $b$. Theorem 3.2 demonstrates that, as the batch size increases, the number of required steps decreases, suggesting that increasing the batch size accelerates DNN training. However, excessively large batch sizes demand substantial computational resources. Therefore, as Theorem 3.3 indicates, the critical batch size that minimizes SFO complexity is significant and practical because moderate batch sizes are suitable for implementing SGD. We have added a discussion of these insights and relationships of Theorems 3.1--3.3 to Section 3.4.
>
>
> **Weakness 2:** The proof of the main result is deferred to the appendix, being a theoretical paper there should have been a proof outline in the main text.
>
> **Requested Changes 2:** It would be good to have a proof outline of the main theorem explaining the key steps of the proof in the main text itself. You can use https://arxiv.org/pdf/1308.6594 for inspiration.
>
> **Reply:** Thank you for this suggestion. We have added a proof outline of Theorem 3.1 to Section 3.1.

---

> ### Author Response · Authors · 2025-02-11
> **Replies for Reviewer qD9t's comments**
>
> **Weakness 4:** The numerical results are not clearly discussed.
>
> **Requested Changes 6:** In the numerical results a discussion on the comparision with the other methods would be nice. The conclusion "Therefore, we can conclude that Algorithm 1 using the critical batch size b performs as well as other optimizers using any batch size in the sense of minimizing the SFO complexities needed to train ResNet18 on the MNIST dataset." does not seem to have enough justification given the results. Figure 7 and 8 are not even discussed. Each numerical result should ideally have a hypothesis associated with it which it either rejects or fails to.
>
>
> **Reply:** Thank you very much for your feedback.
>
> The Armijo condition (see (11)) implies that, if $c$ is large, then $\alpha_k$ satisfying the Armijo condition is small, which implies that SGD with a small $\alpha_k$ would not work. In fact, as indicated in Figure 1, for training ResNet-18 on the CIFAR-10 dataset, when the batch size exceeded $2^7$, SGD using Armijo line search with $c = 0.1$ required a greater number of steps for convergence than with $c = 0.01, 0.001$ because the learning rate at $c=0.1$ was much smaller than that at $c=0.01, 0.001$ (see  Appendix A.8).
> Therefore, the appropriate value for hyperparameter $c$ depends on the specific training dataset. As a result, as shown in Figure 2, SGD using Armijo line search with $c = 0.1$ had a larger SFO complexity than with $c = 0.01, 0.001$ . However, the trend in which smaller values of $c$ resulted in fewer steps and reduced SFO complexity was not observed when training ResNet-18 on the MINIST dataset (Figures 5 and 6). This is because the learning rate increased with the batch size, unlike with the CIFAR-10 dataset. Therefore, an appropriate value for hyperparameter $c$ is needed to implement SGD with Armijo line search and large batch sizes.
>
> A comparison of SGD using the Armijo-line-search learning rate ($c = 0.1$) with baseline optimizers such as SGD, Momentum, Adam, AdamW, and RMSProp (Figures 3 and 4) for training ResNet-18 on the CIFAR-10 dataset reveals that SGD using the Armijo-line-search learning rate performed better than the baselines for any batch size in the sense of minimizing the number of steps and SFO complexity.
> Figures 7 and 8 (training ResNet-18 on MNIST dataset) show that the adaptive methods, Adam and AdamW, performed well. When the batch size was from $2^5$ to $2^8$, SGD performed better than SGD using the Armijo-line-search learning rate in the sense of minimizing the number of steps and SFO complexity. When the batch size was large such as $b = 2^9, 2^{10}$, or $2^{11}$, SGD using the Armijo-line-search learning rate performed better than SGD and Momentum in the sense of minimizing the number of steps and SFO complexity. In short, Figures 3, 4, 7, and 8 show that, when using a large batch size, SGD using Armijo line search performed better than SGD and Momentum.
>
> We have added a discussion of these insights to Section 3.4.
>
>
> **Requested Changes 4:** What is $\nabla f_{\xi_k,k}$ in A2?
>
> **Reply:** (A2) uses $\nabla f_{\xi_{k,i}}$ $(i = 1,2,\cdots,b)$. $\nabla f_{\xi_{k,i}}$ is the gradient of $f_{\xi_{k,i}}$, which is the loss function with a random variable $\xi_{k,i}$ generated by the $i$-th sampling in the $k$-th iteration.
>
>
> **Requested Changes 5:** Can Algorithm 1 and Algorithm 2 be combined? I understand Algorithm 2 is way to do Algorithm 1 but is there a cleaner way?
>
> **Reply:** Thank you for your suggestion. We have combined Algorithms 1 and 2.

---

> > ### Comment · Reviewer_qD9t · 2025-02-14
> > **Thanks for resolving the comments.**
> >
> > Thanks for taking care of most of the comments that I had posted. However I want to discuss a bit further on the implication of your results. Is using as large a batch size as possible always optimal when using Armijo Line Search with SGD as Theorem 3.1 indicates? or are there caveats? what are the limitations here except ofcourse the assumptions on the functions.

---

> > > ### Author Response · Authors · 2025-02-14
> > > **Replies to Reviewer qD9t's comments**
> > >
> > > Thank you for your comments.
> > > Theorem 3.1 indicates that the upper bound of the minimum value of $\mathbb{E}[ \| \nabla f({\theta}_k)\|^2 ]$ consists of a bias term $B({\theta}_0,K)$ and variance term $V(\sigma^2, b)$. When the number of steps $K$ is large and the batch size $b$ is large, $B({\theta}_0,K)$ and $V(\sigma^2, b)$ become small.
> > > Hence, as the reviewer pointed out, using large batch size is optimal in the sense of minimizing the minimum value of $\mathbb{E}[ \| \nabla f({\theta}_k)\|^2 ]$ under our assumptions.
> > > However, excessively large batch sizes demand substantial computational
> > > resources. Therefore, as Theorem 3.3 indicates, the critical batch size that minimizes the SFO complexity, which is the stochastic gradient computation cost, would be useful to implement SGD with the Armijo line search. In addition, our numerical results showed that moderate batch sizes around the critical batch size are suitable for implementing SGD using the Armijo line search.
> > >
> > > The limitations of our theorems are such that, while Theorem 3.1 shows in theory that using large batch sizes minimizes the minimum value of $\mathbb{E}[ \| \nabla f({\theta}_k)\|^2 ]$,  using excessively large batch sizes would be unrealistic in practice, since they demand  substantial computational resources.
> > > Hence, this paper (Theorem 3.3) suggests using the critical batch size that minimizes the SFO complexity to implement SGD with the Armijo line search.
> > > Our theorems guarantee an $\epsilon$-approximation of SGD with both the Armijo line-search-learning rate and the critical batch size.

---

> > > > ### Comment · Reviewer_qD9t · 2025-03-01
> > > > **Thanks for the response**
> > > >
> > > > Thanks for replying to my query, I have submitted my official recommendation.

---

### Review · Reviewer_wtqH · 2025-01-28

**Summary Of Contributions:**

The paper considers the convergence of stochastic gradient descent with minibatch and Armijo search. The main result is a convergence rate of SGD for nonconvex problems under a smoothness assumption. The paper shows that the Armijo search implies rates within a bounded interval. Based on this, the paper shows that the total iteration complexity is a convex function of the batch size, from which the paper shows a principle to set the batch size.

**Audience:**

Yes

**Broader Impact Concerns:**

not applicable.

**Claims And Evidence:**

Yes

**Requested Changes:**

I would like the authors to present changes to address my comments above, especially about the correctness of the proof.

**Strengths And Weaknesses:**

**Strength**

- The Armijo line search is an effective approach to automatically determining the step size. The paper gives some interesting results on the properties of Armijo line search such as the bound on the step size found by the Armijo search.
- The paper also gives a result to show the total iteration complexity as a function of the batch size. This implies a principled way to determine the batch size.
- The paper includes both theoretical analysis and empirical verification.

**Weakness**
- The main result on the convergence rates involve both the upper bound and lower bound of the step size, e.g., $\bar{\alpha}$ and $\underline{\alpha}$. Note that the upper bound may be much larger than the actual step size used in the algorithm, and the lower bound may be much slower. In this case, the convergence rate in Theorem 3.1 may be very slow. For example, the authors consider the case $\bar{\alpha}>1/L_n$, which is a constant independent of the number of iterations. However, in practical SGD uses a much slower step size in practice, e.g., a typical choice is $1/\sqrt{K}$ in SGD with constant step size. Furthermore, the paper use $\bar{\alpha}=10$ in Section 4. In this case, the upper bound and the actual step size found by the algorithm may be significant, and the resulting convergence rate may be not quite effective.
- In the inequality above Eq (22), the upper bound involves $(L_n\alpha_k-1)\alpha_k\|\nabla f_{B_k}(\theta_k)\|^2$. In Eq (27), the paper bounds this term by $(L_n\bar{\alpha}-1)\bar{\alpha}\|\nabla f_{B_k}(\theta_k)\|^2$. However, it seems that this inequality is not correct. Actually, the function $\alpha\mapsto (L_n\alpha-1)\alpha$ is decreasing if $\alpha>1/(2L_n)$. Therefore, this inequality does not hold if $\alpha_k>1/(2L_n)$.
- Eq (25) is not rigorous. First, $-(L_n\bar{\alpha}-1)\underline{\alpha}$ should be $(L_n\bar{\alpha}-1)\underline{\alpha}$. Second, the identity does not hold since $(L_n\bar{\alpha}-1)\underline{\alpha}+\bar{\alpha}$ is not equal to $L_n\bar{\alpha}^2$.

---

> ### Author Response · Authors · 2025-02-11
> **Replies for Reviewer wtqH's comments**
>
> We appreciate your detailed assessments and helpful feedback. We have revised the manuscript to incorporate all of the recommendations, which has resulted in an improved presentation of our work. The revised parts of the manuscript are marked in red.
>
> **Weakness 1:** The main result on the convergence rates involve both the upper bound and lower bound of the step size, e.g., $\overline{\alpha}$ and $\underline{\alpha}$. Note that the upper bound may be much larger than the actual step size used in the algorithm, and the lower bound may be much slower. In this case, the convergence rate in Theorem 3.1 may be very slow. For example, the authors consider the case $\overline{\alpha} > 1/L_n$, which is a constant independent of the number of iterations. However, in practical SGD uses a much slower step size in practice, e.g., a typical choice is $1/\sqrt{K}$ in SGD with constant step size. Furthermore, the paper use $\overline{\alpha}=10$ in Section 4. In this case, the upper bound and the actual step size found by the algorithm may be significant, and the resulting convergence rate may be not quite effective.
>
> **Reply:** Thank you very much for your feedback. In the experiment (described in Section 4), we set upper bound $\overline{\alpha}=10$. Then, SGD with Armijo line search sometimes used $\alpha_k =10$, as nowexplained in Appendix A.8. However, since the lower bound $\underline{\alpha}$ of $\alpha_k$ involves an unknown constant ($L_n$), the effect of using $\alpha_k =10$ is unclear.
>
> We also compared SGD using Armijo line search with SGD using $\alpha_k = 1/\sqrt{K}$ for training ResNet-18 on the CIFAR-10 dataset. Batch size $b$ was $128$, and the number of epochs $E$ was $120$ (which implies that $K = 46920$). The comparison indicates that using Armijo line search results in better performance than using constant learning rate $\alpha_k = 1/\sqrt{K}$, as explained in Appendix A.9.
>
> **Weakness 2:** In the inequality above Eq (22), the upper bound involves $(L_n \alpha_{k-1} -1)\alpha_k |\nabla f_{B_k}({\theta}\_k)|^2$. In Eq (27), the paper bounds this term by $(L_n \bar{\alpha} -1) \bar{\alpha} |\nabla f_{B_k}({\theta}_k)|^2$. However, it seems that this inequality is not correct. Actually, the function $\alpha \mapsto (L_n \alpha - 1) \alpha$ is decreasing if $\alpha > 1/(2L_n)$. Therefore, this inequality does not hold if $\alpha_k > 1/(2L_n)$.
>
> **Reply:** Thank you very much for your feedback. The function $g \colon \alpha\mapsto (L_n\alpha-1)\alpha = L_n(\alpha -\frac{1}{2L_n})^2-\frac{1}{4L_n}$ increases for $\alpha >1/(2L_n)$ (for example, $g(1/(2 L_n)) = - 1/(4 L_n)$ and $g(1/L_n) = 0$).
>
> **Weakness 3:** Eq (25) is not rigorous. First, $-(L_n \overline{\alpha}-1)\underline{\alpha}$ should be $(L_n \overline{\alpha}-1)\underline{\alpha} $. Second, the identity does not hold since $(L_n\overline{\alpha}-1)\underline{\alpha}+\overline{\alpha}$ is not equal to $L_n\overline{\alpha}^2$.
>
> **Reply:** Thank you very much for your feedback. We have modified (25) as follows:
>
> \begin{align*}
> \mathbb{E}_{\xi\_k} \left[f({\theta}\_{k+1}) | {\theta}\_k \right]&\leq f({\theta}\_{k})-\frac{\underline{\alpha}}{2} \|\nabla f({\theta}\_k)\|^2+\frac{1}{2} (L_n \overline{\alpha} - 1)\underline{\alpha}\left(\|\nabla f({\theta}\_k) \|^2 + \frac{\sigma^2}{b}\right)+ \frac{\overline{\alpha} \sigma^2}{2b}
> \end{align*}
>
> \begin{align*}
> &=f({\theta}\_{k})-\frac{\underline{\alpha}}{2} \|\nabla f({\theta}\_k)\|^2+\left( \frac{(L_n \overline{\alpha} -1) \underline{\alpha}}{2} \right) \|\nabla f({\theta}\_{k})\|^2+ \frac{ ((L_n \overline{\alpha}-1)\underline{\alpha}+\overline{\alpha}) \sigma^2}{2b}.
> \end{align*}
> Theorem 3.1(i) is now therefore modified, from
>
> \begin{align*}
> \min\_{k \in [0:K-1]} \mathbb{E}\left[ \| \nabla f({\theta}\_k)\|^2 \right] \leq \frac{2(f({\theta}\_{0}) - f_*)}{\tilde{\alpha} - (L_n \overline{\alpha} -1) \overline{\alpha}}\frac{1}{K}+ \frac{L_n \overline{\alpha}^2 \sigma^2}{\tilde{\alpha} - (L_n \overline{\alpha} -1) \overline{\alpha}}\frac{1}{b}
> \end{align*}
>
> to
>
> \begin{align*}
> \min\_{k \in [0:K-1]} \mathbb{E}\left[ \| \nabla f({\theta}\_k)\|^2 \right]\leq \frac{2(f({\theta}\_{0}) - f_*)}{\tilde{\alpha} - (L_n \overline{\alpha} -1) \overline{\alpha}}\frac{1}{K}+ \frac{\{(L_n \overline{\alpha}-1)\underline{\alpha}+\overline{\alpha}\} \sigma^2}{\tilde{\alpha} - (L_n \overline{\alpha} -1) \overline{\alpha}} \frac{1}{b}.
> \end{align*}
>
> We have changed the relevant proofs in the main text as shown by the red text.

---

### Comment · Action_Editor_Cc3W · 2025-02-14
**Author's response and revised manuscript now available**

Dear reviewers,

Please take a look at the author's response and the revised manuscript under the revisions tab. Can you please comment on whether the rebuttal and revision resolves your concerns from the previous-round submission?

---

### Decision · Action_Editor_Cc3W · 2025-03-06

**Recommendation:** Accept as is

**Comment:**

See above.

**Audience:**

Reviewer 1 states: ``I believe the paper should be accepted because it does study a relevant optimization algorithm and analyze the convergence with respect to batch size and number of steps"

Reviewer 2 states "he paper presents interesting observations on the setting of the batch size to speed up the convergence."

Reviewer 3 states "authors have addressed my concerns"

**Claims And Evidence:**

The authors have resolved all the concerns of the reviewers that arise from previous submission rounds. I concur with this assessment.